

# Nutrient availability and the ultimate control of the biological carbon pump in the Western Tropical South Pacific Ocean

Thierry Moutin[1], Thibaut Wagener[1], Mathieu Caffin[1], Alain Fumenia[1], Audrey Gimenez[1], Melika Baklouti[1], Pascale Bouruet-Aubertot[2], Mireille Pujo-Pay[3], Karine Leblanc[1], Dominique Lefevre[1], Sandra
Helias Nunige[1], Nathalie Leblond[4], Olivier Grosso[1] and Alain de Verneil[1]

[1] Aix Marseille Université, CNRS, Université de Toulon, IRD, OSU Pythéas, Mediterranean Institute of Oceanography (MIO), UM 110, 13288, Marseille, France.
[2] Sorbonne Universités - UPMC Univ. Paris 06 - LOCEAN, BP100, 4 place Jussieu, 75252 Paris cedex 05, France.
[3] Laboratoire d'Océanographie Microbienne – UMR 7321, CNRS - Sorbonne Universités, UPMC Univ Paris 06, Observatoire
Océanologique, 66650 Banyuls-sur-mer, France.
[4] Observatoire Océanologique de Villefranche, Laboratoire d'Océanographie de Villefranche, UMR 7093, Villefranche-sur-mer, France.

*Correspondence to*: Thierry Moutin (Thierry.moutin@mio.osupytheas.fr)

**Abstract**

Surface waters (0-200 m) of the western tropical South Pacific (WTSP) were sampled along a longitudinal 4000 km transect (OUTPACE cruise, 18 Feb., 3 Apr. 2015) during the stratified period between the Melanesian Archipelago (MA) and the western part of the SP gyre (WGY). Two distinct areas were considered for the MA, the western MA (WMA) and the eastern MA (EMA). The main carbon (C), nitrogen (N), phosphorus (P) pools and fluxes allow for characterization of the expected
trend from oligotrophy to ultra-oligotrophy, and to build first-order budgets at the daily and seasonal scales (using climatology). Sea surface chlorophyll a reflected well the expected oligotrophic gradient with higher values obtained at WMA, lower values at WGY and intermediate values at EMA. As expected, the euphotic zone depth, the deep chlorophyll maximum and nitracline depth deepen from west to east. Nevertheless, phosphaclines and nitraclines did not match. The decoupling between phosphacline and nitracline depths in the MA allows excess P to be locally provided in the upper water by winter
mixing. We found a significant biological "soft tissue" carbon pump in the MA sustained almost exclusively by $N_2$ fixation and essentially controlled by phosphate availability in this iron-replete environment. The MA appears to be a net sink for atmospheric $CO_2$ while the WGY is in quasi steady state. We suggest that the necessary excess P, allowing the success of nitrogen fixers and subsequent carbon production and export, is mainly brought to the upper surface by local deep winter convection at an annual scale rather than by surface circulation. We also suggest that mesozooplankton diel vertical migration
plays a dominant role in the transfer of carbon from the upper surface to deeper water in the MA. While the origin of the decoupling between phosphacline and nitracline remains uncertain, the direct link between local P upper waters enrichment, $N_2$ fixation, organic carbon production and export, offers a possible shorter time scale than previously thought between N input by $N_2$ fixation and carbon export. The low iron availability in the SP gyre and P availability in the MA during the stratified period may appear as the ultimate control of N input by $N_2$ fixation. Because of the huge volume of water to consider and
because the SP Ocean is the place of intense denitrification in the east (N sink) and $N_2$ fixation in the west (N source), precise seasonal C, N, P budgets would be of prime interest to understand the efficiency, at the present time, and in the future, of the oceanic biological carbon pump.

## 1 Introduction

The oceanic biological carbon pump corresponds to the transfer of carbon from the upper surface to the ocean interior by
biological processes, greatly influencing atmospheric $CO_2$ concentration and therefore the earth's climate. It is a high rank priority of contemporaneous research in oceanography (Burd et al., 2016). Two biological pumps have been defined (Volk and

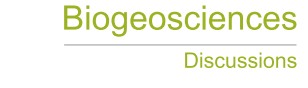

Hoffert, 1985), the "soft tissue" and "carbonate" pumps associated with organic matter or calcium carbonate processes (e.g. production, export, remineralization or dissolution). The "soft tissue" pump, considering both its intensity and shorter time scales, is by far the larger contributor to the mineral carbon gradient between the upper surface and the deep Sea. Following climate alteration, the biological "soft tissue" pump begins to deviate from its equilibrium condition, meaning that its influence

on atmospheric $CO_2$ change may occur at time scales shorter than previously thought (Sarmiento and Grüber, 2006). Because the strength of the biological carbon pump depends on nutrient availability in the upper ocean, and more particularly on nitrogen availability (Falkowski et al., 1998, Tyrell, 1999, Moore et al., 2013), which is at long term regulated by external input by dinitrogen ($N_2$) fixation and internal denitrification (Grüber and Sarmiento, 1997; Codispoti et al., 2001; Deutsch et al., 2001; Brandes and Devol, 2002; Grüber, 2004; Mahaffey et al., 2005; Deutsch et al., 2007; Codispoti et al., 2007; Capone

and Knapp, 2007; Moutin et al., 2008; Deutsch and Weber, 2012; Landolfi et al., 2013, Jickells et al., 2017), quantitative evaluation of the regulation, interdependence, and evolution of these two processes requires intense attention at the present time. It has been suggested earlier that $N_2$ fixation may play a large part in changing atmospheric $CO_2$ inventories (McElroy et al., 1983), but at long time scales and considering large differences in Aeolian iron input (Falkowski, 1997, Broecker and Henderson, 1998). Because $N_2$ fixation may ultimately be controlled by iron availability, and because dust delivery to the

ocean is climate sensitive, there may be inextricably linked feedback mechanisms that regulate $N_2$ fixation, atmospheric $CO_2$ concentrations, and dust deposition over relatively long periods (Michaels et al., 2001; Karl, 2014). Although fundamental, the time scales by which N sources and sinks are coupled in the ocean remain uncertain (Falkowski et al., 1998; Brandes and Devol, 2002; Straub et al., 2013). Excess P emerges as a master variable to link them in the modern ocean (Deutsch et al., 2007) as well as from a paleobiogeochemical point of view (Straub et al., 2013). The recent (since the beginning of the

industrial era) increase in production by $N_2$-fixing cyanobacteria was suggested to provide a negative feedback to rising atmospheric carbon dioxide concentrations (McMahon et al., 2015) although an inverse trend was also proposed (Kim et al., 2017). While the observed changes in $N_2$ fixation and biogeochemical cycling reflect either natural oceanic variability or climate change (Karl et al., 1997; Karl, 2014), the most probable changes for the near future in both $N_2$ fixation and denitrification processes following climate forcing are suggested to be a strengthening control of the carbon cycle by P

availability (Moutin et al., 2008).

The western tropical South Pacific (WTSP) is a poorly studied area where large blooms of diazotrophs were previously observed by satellite (Dupouy et al., 2000; 2011) and has been recently qualified as a hot spot of $N_2$ fixation (Bonnet et al., 2017). It is hypothesized that following the South Equatorial Current (SEC), the N-depleted, P-enriched waters from areas of denitrification located in the East Pacific reach waters with sufficient iron in the west to allow $N_2$ fixation to occur (Moutin et

al., 2008; Bonnet et al., 2017). While horizontal advection of waters from the east through the SEC probably supports an active biological pump in the WTSP, local vertical convection may also play a central role.

In addition to the main objective of following the same water mass for several days (de Verneil et al., 2017) by a quasi-Lagrangian experiment (Moutin et al., 2017) in order to propose daily budgets (Caffin et al., this issue; Knapp et al., this issue) or short term biological trends (Van Wambeke et al., this issue), we proposed here to work at larger space and time scales, in

complement to the work by Fumenia et al. (this issue) showing that $N_2$ fixation in the WTSP may influence the whole SP Ocean. While many recent works focus on important small spatial scales influencing the biological carbon pump (Lévy et al., 2012; Stukel et al., 2017), we found it important to also show results from a larger scale study in the OUTPACE (Oligotrophy to UlTra-oligotrophy PACific Experiment) special issue (https://www.biogeosciences.net/special_issue894.html), showing that they are complementary rather than exclusive. This study was also motivated because we are far from resolving seasonal

variations of the main biogeochemical variables in the WTSP, still largely under-sampled compared to the northern Pacific and Atlantic. The objective of this study is therefore to provide a large spatial (hundreds of km) and temporal (annual) scale study of the main biogeochemical C, N, P stocks and fluxes in the upper 200 m of the WTSP Ocean from measurements





gathered during the stratified period and to evaluate the main seasonal trends from estimations of previous winter conditions and climatological analysis.

**2 Material and methods**

**2.1 General method and strategy**

*Station locations, chronology, CTD measurements, sample collection*

The OUTPACE cruise was carried out between 18 February and 3 April 2015 from Nouméa (New Caledonia) to Papeete (French Polynesia) in the WTSP (Fig. 1). We sampled water along a 4000 km transect from the oligotrophic water of the MA to the clearest ocean waters of the South Pacific (SP) gyre (Moutin et al., 2017) from a SBE 911+ CTD-Rosette. Euphotic zone depth (EZD) was immediately determined on-board from the photosynthetic available radiation (PAR) in depth compared to

the sea surface PAR($0^+$), and used to determine the upper waters sampling depths corresponding to 75, 54, 36, 19, 10, 3, 1 (EZD), 0.3, and 0.1 % of PAR($0^+$). CTD sensors were calibrated and data processed post-cruise using Sea-Bird Electronics software into 1-m bins. Conservative temperature, absolute salinity and potential density were computed using TEOS-10 (McDougall and Barker, 2011). Chlorophyll a (chl a) in mg m$^{-3}$ were measured with an Aqua Trak III fluorimeter (Chelsea Technologies Group Ltd). All samples were collected from the 24, 12-L Niskin bottles equipped with silicone rubber closures

and tubing for measurements (see section 2.2. Analytical method) of stock variables (dissolved oxygen, dissolved inorganic carbon (DIC), total alkalinity (TA), nutrients, chl a, particulate and dissolved organic C, N, P) and fluxes (primary and bacterial production rates, N$_2$ fixation rates, and dissolved inorganic phosphate (DIP) turnover times, i.e. the ratio of DIP concentration to DIP uptake).

*Group of stations*

For our large-scale study, we considered 3 areas: the western MA (WMA), the eastern MA (EMA) and the western gyre (WGY) waters. Four 0-200 m CTD casts, mainly devoted to nutrient pool analyses, were considered for each area and correspond to the following stations: SD 1, SD 2, SD 3 and LD A for WMA, SD 6, SD 7, SD 9 and SD 10 for EMA and SD 13, SD 14, SD 15 and LD C for WGY (Fig. 1, Tables 1 & 2). Therefore, the same number of CTD casts was used to characterize

each area. The choice of the stations for each area was essentially geographic but justified *a posteriori* by the results. SD 8 was discarded because no nutrient measurements were available. SD 11, SD 12 and LD B were also discarded because a bloom was sampled at LD B, meaning these measurements are out of the scope of this paper, which deals with large-scale spatial and temporal variations. The specificity of the transition area between the MA and GY waters are presented in another paper of the OUTPACE special issue (de Verneil et al., 2017). WMA, EMA and WGY will be presented in dark green, light green and

blue, respectively, in close relationship with the expected oligotrophic gradient.

*Mixed layer depths*

Mixed layer depth (MLD) was calculated using a threshold temperature of 0.2 °C deviation from the reference value at 10-m depth (de Boyer-Montegut et al., 2004) from OUTPACE CTD profiles (Table 1). For climatological MLD data (Fig. 2a, 2d,

2g), values at each station were extracted from the global climatology at 2° resolution proposed by de Boyer-Montegut et al. (2004) (downloaded from http://www.ifremer.fr/cerweb/deboyer/mld/Surface_Mixed_Layer_Depth.php on Jan 12, 2017). The same criterion (threshold temperature deviation of 0.2 °C) was used.

*Vertical eddy diffusivity measurement*

The mean eddy vertical diffusivity between 40-200 m was determined for each station from one to several casts undertaken using a VMP1000 (Bouruet-Aubertot et al., this issue). Briefly, Kz is inferred from the dissipation rate of turbulent kinetic



energy, ε, mixing efficiency, γ, and buoyancy frequency, N, according to the Osborn relationship: $K_z=(\gamma\varepsilon)/N^2$. ε is computed from the microstructure shear measurements (e.g. Xie et al, 2013) and mixing efficiency is inferred from the Bouffard and Boegman parameterization as a function of turbulence intensity (Bouffard and Boegman, 2013).

*Satellite data*

Sea surface temperature (SST) (Fig. 2b, 2e, 2h) and sea surface chl a (SSchl a) (Fig. 2c, 2f, 2i) from July 2014 to July 2015 were obtained using processed satellite data provided by the MODIS Aqua mission (downloaded from https://oceandata.sci.gsfc.nasa.gov/ on Jan 3, 2017). The mapped level 3 reanalysis has a 4 km spatial resolution produced at a monthly time scale. For each station, pixels within a rectangle with sides +/- 1/8° longitude and latitude away from the station

position were averaged together to produce a single value.

*Depth profiles of all discrete variables*

All measurements are presented together with their estimated mean concentrations profile (thick line) on Figs. 3, 4, 5, 6. In order to determine the mean concentrations, the profiles of the variable in question (concentration vs depth) for all stations

included in the group were interpolated between 5 and 200 m with a piecewise cubic hermite interpolating scheme (pchip function in the pracma R package). In case of missing values close to 200 m, the interpolation was stopped at the deepest (before 200 m) point available. The mean profile was estimated from the mean value of the interpolated profiles on every one meter depth horizon. For inorganic nutrient concentrations < quantification limit (QL) (see section 2.2), a zero was indicated in order to show that a measurement was taken.

*Normalization*

Concentrations normalized by salinity are used to study biological processes independent of variations related to evaporation/precipitation. At global scales, it is common to apply $S_P=35$ (Millero, 2007). In order to estimate seasonal trends in our specific areas, we normalized to the mean absolute salinity measured at 70 m depth in each area, $S_A = 35.65 \pm 0.04$,

$35.83 \pm 0.04$ and $35.91 \pm 0.02$ g.kg$^{-1}$ for the WMA, EMA and WGY, respectively. This choice will be further justified hereafter. Important differences in the carbonate system require to take into account this normalization, which justifies its use for the other variables, even if changes are relatively small (e.g., for nutrients).

*Inventories*

Inventories were calculated from the depth profiles of the discrete variables of inorganic and organic C, N, P dissolved and particulate pools (see section 2.2) measured during the OUTPACE cruise (Table 3) between 0 and 70 m depth. The latter depth corresponded to the average deeper annual MLD obtained using climatology as explained above and shown in Fig. 2 (a, d, g). The integrated fluxes were calculated considering the same depths.

*Settling particulate matter and swimmers mass, C, N and P flux measurements*

The settling of particles in the water column outside of the upper layer was measured using 2 PPS5 sediment traps (1 m$^2$ surface collection, Technicap, France) deployed for 4 days at 150 and 330 m at LD A (MA) and LD C (WGY) stations (Fig. 1). Particle export was recovered in polyethylene flasks screwed on a rotary disk which allowed automatically changing the flask every 24-h to obtain a daily material recovery. The flasks were previously filled with a buffered solution of formaldehyde

(final conc. 2 %) and were stored at 4 °C after collection until analysis to prevent degradation of the collected material. Onshore, swimmers were handpicked from each sample. Settling particulate matter and swimmers were both weighted and analyzed separately on Elemental Analyzer coupled to an Isotope Ratio Mass Spectrometer EA-IRMS (Integra2, Sercon Ltd)



to quantify total C and N. Total P was analyzed as described in section 2.2. The total element measurements for the settling particulate matter were considered to represent the settling particulate organic C, N, P. The results are presented Table 4.

*Ocean-atmosphere $CO_2$ fluxes*

Ocean-atmosphere $CO_2$ fluxes $\Phi_{CO2} = -k_g * (p_{CO2}{}^{atm} - p_{CO2}{}^{oc})$ were calculated considering 1) a mean $k_g$ of $0.031 \pm 0.005$ mol m$^{-2}$ y$^{-1}$ µatm$^{-1}$ (i.e. 85 µmol m$^{-2}$ d$^{-1}$ µatm$^{-1}$) for gas transfer velocity estimated from Liss and Melivat (1986) relationship and sea winds derived from satellite measurements (1999-2009). Data came from Boutin et al. (downloaded from http://cersat.ifremer.fr/ on March 3 , 2017 and extracted on a geographical grid (Latitude : -17 to -23° N, Longitude: +159 to +211° E) - one grid was used because no significant differences were obtained in $k_g$ for the 3 areas WMA, EMA and WGY), 

2) a mean oceanic $p_{CO2}$ ($p_{CO2}{}^{oc}$) determined for each area during the OUTPACE cruise and 3) a mean atmospheric $p_{CO2}$ ($p_{CO2}{}^{atm}$) estimated from the molar fraction of $CO_2$ ($X_{CO2}$) in dry air measured at SMO station Tutuila (American Samoa, Lat 14.247° S, Lon 170.564° W, north of LD B (Fig. 1), NOAA/ESRL - data downloaded from http://dx.doi.org/10.7289/V51834DB on February 7 , 2017). A monthly averaged $X_{CO2} = 398.4$ ppm for March 2015 was used whereas $X_{CO2}$ varied from 396.0 to 398.4 ppm from July 2014 to July 2015 at Tutuila with an annual mean of 397.3 ppm. The $X_{CO2}$ data were converted in $p_{CO2}{}^{atm}$

considering 100% humidity and a total pressure of 1 atm (101325 Pa) following Weiss and Price (1980) with surface seawater temperature and salinity of each area (Table 5). A total pressure of $101260 \pm 180$ Pa was determined considering NCEP-NCAR Reanalysis 1 on the OUTPACE area from July 2014 to July 2015, with no longitudinal trend, justifying to consider 1 atm as total pressure for the conversion (downloaded from https://www.esrl.noaa.gov/psd/data/gridded/data.ncep.reanalysis.html on December 19, 2017).

*Upper layer (0-70 m) daily C, N, P, budgets*

Comparative daily C, N, and P budgets of the upper 70 m layer were established for each area (Table 6). Inputs from below associated with vertical turbulent diffusion were calculated using the mean vertical eddy diffusivity, and slopes of nutriclines (Table 2) and DIC gradients calculated between 70-200 m using linear regressions (data not shown). The ocean-atmosphere 

$CO_2$ fluxes were detailed in the previous paragraph. The input of nitrogen by $N_2$ fixation was calculated for each area (Table 6) using depth profile sampling and on-deck 24-h $^{15}N_2$ incubations (section 2.2). Both C, N, P particulate and dissolved organic export were estimated. The way to obtain particulate export by settling material (Table 4) was described above. Output of dissolved and particulate organic matter by turbulent diffusion was calculated from the mean vertical eddy diffusivity (Table 1) and from gradients estimated with linear regressions (data not shown) between the surface and 70 m depth of DOC-POC 

(Fig. 5d-5g), DON-PON (Fig. 5e-5h) and DOP-POP (Fig. 5f-5i). When non-significant gradients were obtained, fluxes were nil.

*Seasonal variations and upper layer (0-70 m) annual C, N, P budgets*

We sampled for OUTPACE during the stratified period characterized by minimum MLDs close to 20-40 m (Fig. 2a, 2d, 2g), where the largest part of biological fluxes (Fig. 6) occurred. Because the only mechanism able to disrupt this stratification at a large scale is deep water mixing occurring during winter, and more specifically in July in this area (Fig. 2a, 2d, 2g), we postulated that conditions at 70 m depth (average depth of wintertime MLD) remained unchanged, or did not significantly change, all over the year. Considering no large inter-annual differences in winter MLDs, we considered that the mean 

measurements at 70 m depth during OUTPACE well represented the homogeneous upper water column (0-70 m) variables and initial winter conditions (i.e. conditions in July 2014) allowing to draw first-order winter to summer seasonal variations (Table 7) and 8-month C, N, P budgets (Table 8). The dashed lines in all Figs. 3, 4 and 5 indicate the upper surface expected





values for all variables during the 2014 austral winter, and allow for evaluation of the temporal variation toward the austral
summer season (full lines) in each area.

*Surface waters carbonate system climatology*

The climatological gridded values proposed in Takahashi et al. (2014), hereafter referred as NDP-094 climatology, were used
to validate our estimated values for the carbonate system in the upper surface previous winter conditions (July 2014). The
dataset is based on interpolated $p_{CO_2}^{OC}$ and calculated TA data (based on regional linear potential alkalinity-salinity
relationships) on a 4° Latitude by 5° Longitude monthly grid in the reference year 2005. The variable DIC (among others) is
calculated from $p_{CO_2}^{OC}$ and TA. Data were downloaded from http://cdiac.ess-dive.lbl.gov/ftp/oceans/NDP_094/ on December
19, 2017. Climatological July data centred on 20°S were extracted along the cruise transect and 2, 3 and 3 pixels were averaged
for comparison in the WMA, EMA and WGY areas, respectively (Table 5). In order to account for the $p_{CO_2}$ increase at the
earth surface between 2005 and 2015, a constant offset of 1.5 $\mu atm.y^{-1}$ was applied to $p_{CO_2}$ and a corresponding constant offset
of 1 $\mu mol\ kg^{-1}\ y^{-1}$ was also applied to DIC.

## 2.2 Analytical chemical methods

*Oxygen and apparent oxygen utilization (AOU)*

Oxygen concentration in the water column was measured with a Seabird SBE43 electrochemical sensor interfaced with the
CTD unit. The raw signal was converted to an oxygen concentration with 13 calibration coefficients. The method is based on
the Owens and Millard (1985) algorithm that has been slightly adapted by Seabird in the data treatment software using a
hysteresis correction. A new set of calibration coefficients has been determined after the cruise to post-process the whole
dataset. Only three coefficients (the oxygen signal slope, the voltage at zero oxygen signal, the pressure correction factor)
among the 13 determined by the pre-cruise factory calibration of the sensor were adjusted with the following procedure: The
oxygen concentrations measured by Winkler were matched with the signal measured by the sensor at the closing of the Niskin
bottles. The three values were fitted by minimizing the sum of the square of the difference between Winkler oxygen and
oxygen derived from sensor signal. Winkler oxygen concentration was measured following the Winkler method (Winkler,
1888) with potentiometric endpoint detection (Oudot et al., 1988) on discrete samples collected with Niskin bottles. For
sampling, reagents preparation and analysis, the recommendations from Langdon (2010) have been carefully followed. The
Thiosulfate solution was calibrated by titrating it against a potassium iodate certified standard solution of 0.0100N (WAKO).
AOU was computed with oxygen concentration at saturation estimated following the algorithm proposed by Garcia and Gordon
(1992) considering Benson and Krause values.

*TA, DIC and $p_{CO_2}^{oc}$*

Samples for total alkalinity (TA) and dissolved inorganic carbon (DIC) were collected from Niskin bottles in one 500 mL glass
flask (Schott Duran) and poisoned directly after collection with $HgCl_2$ (final concentration 20 mg.$L^{-1}$). Samples were stored at
4°C during transport and analyzed 5 months after the end of the cruise at the SNAPO-CO$_2$ (Service National d'Analyse des
paramètres Océaniques du CO$_2$- LOCEAN – Paris). TA and DIC were measured on the same sample based on one
potentiometric titration in a closed-cell (Edmond, 1970). A non-linear curve fitting approach was used to estimate TA and DIC
(Dickson 1981, DOE 1994). Measurements were calibrated with reference materials (CRM) for oceanic CO$_2$ measurements
purchased by the SNAPO-CO$_2$ to Pr. A. Dickson (Oceanic Carbon Dioxide Quality Control, USA). The reproducibility
expressed as the standard deviation of the CRM analysis was 4.6 $\mu mol\ kg^{-1}$ for TA and 4.7 $\mu mol\ kg^{-1}$ for DIC. Moreover, the
standard deviation on the analysis of 12 replicates collected at the same depth (25 m) at station LD C was 3.6 $\mu mol\ kg^{-1}$ for
TA and 3.7 $\mu mol\ kg^{-1}$ for DIC. The Estimation of $p_{CO_2}^{oc}$ was made with the SEACARB R package [Gattuso and Lavigne,



2009]. The dissociation constants $K_1$ and $K_2$ (for carbonates in seawater) from Lueker et al. (2000) were used. When available, phosphate and silicate concentrations were used in the calculation.

*-Nutrient, dissolved and particulate C, N, P pools*

Total C, N, P (TC,TN,TP) in seawater samples may be separated in three pools: the dissolved inorganic C, N, P pools (DIC, DIN, DIP), the dissolved organic C, N, P pools (DOC, DON, DOP) and the particulate organic C, N, P pools (POC, PON, POP). No significant particulate inorganic pools are generally considered in open ocean waters.

Two samples for dissolved inorganic nutrient pools measurements were collected from Niskin bottles in 20-mL Polyethylene bottles and one sample was directly analyzed on-board and the other poisoned with 50µl $HgCl_2$ (20 g.L$^{-1}$) and stored for

analysis after the cruise in the laboratory. DIN = $[NO_3^-]$ + $[NO_2^-]$ + $[NH_4^+]$, sum of nitrate, nitrite and ammonium, respectively. Because $[NO_2^-]$ and $[NH_4^+]$ were negligible compared to $[NO_3^-]$, DIN = $[NO_3^-]$. DIP = $[HPO_4^{2-}]$ + $[PO_4^{3-}]$ = orthophosphates also symbolized as $PO_4$. Nitrate, nitrite and orthophosphates concentrations were determined on a segmented flow analyzer (AAIII HR SEAL ANALYTICAL) according to Aminot and Kérouel (2007) with a QL of 0.05 µmol L$^{-1}$. Ammonium was measured by fluorometry (Holmes et al., 1999; Taylor et al., 2007) on a fluorimeter Jasco FP-2020 with a QL of 0.01 µmol L$^{-1}$

15   .

The dissolved organic pools, DON and DOP, were measured using high-temperature (120 °C) persulfate wet-oxidation mineralization (Pujo-Pay and Raimbault, 1994). Samples were collected from Niskin bottles in 100 mL combusted glass bottles and immediately filtered through 2 pre-combusted (24h, 450 °C) glass fiber filters (Whatman GF/F, 25mm). Filtered samples were then collected in Teflon vials adjusted at 20 mL for wet oxidation. Nitrate and phosphate formed, corresponding to total

dissolved pool (TDN and TDP) were then determined as previously described for the dissolved inorganic pools. DON and DOP were obtained by difference between TDN and DIN, and TDP and DIP, respectively. The precision and accuracy of the estimates decreased with increasing depth, as inorganic concentrations became the dominant component in the total dissolved nutrient pools. The limits of quantification were 0.5 and 0.05 µmol L$^{-1}$ for DON and DOP, respectively. The same pre-filtration was used for dissolved organic carbon (DOC) measurements. Filtered samples were collected into glass pre-combusted

ampoules that were sealed immediately after samples were acidified with orthophosphoric acid ($H_3PO_4$) and analyzed by high temperature catalytic oxidation (HTCO) (Sugimura and Suzuki, 1988; Cauwet, 1994, 1999) on a Shimadzu TOC-L analyzer. Typical analytical precision is ± 0.1–0.5 (SD). Consensus reference materials (http://www.rsmas.miami.edu/groups/biogeochem/CRM.html) was injected every 12 to 17 samples to insure stable operating conditions.

The particulate pools (PON, POP) were determined using the same wet oxidation method (Pujo-Pay and Raimbault; 1994). 1.2-L samples were collected from Niskin bottles in polycarbonate bottles and directly filtered onto a pre-combusted (450 °C, 4 h) glass fiber filter (Whatman 47 mm GF/F). Filters were then introduced in teflon vials with 20 mL of ultrapure water (Milli-Q grade) and 2.5 mL of wet oxidation reagent for mineralization. Nitrate and orthophosphates produced were analyzed as described before. QLs are 0.02 µmol L$^{-1}$ and 0.001 µmol L$^{-1}$ for PON and POP, respectively. Particulate organic carbon (POC)

was measured using a CHN analyzer and the improved analysis proposed by Sharp (1974).

*Primary production rates and DIP turnover times*

Vertical profiles of DIC uptake ($V_{DIC}$) and phosphate turnover time ($T_{DIP}$) have been measured once at each station using a dual-labeling method ($^{14}C$ and $^{33}P$) considering a $^{33}P$ period $T_{1/2}$ = 25.55 ± 0.05 days (Duhamel et al., 2006). Each sample (150-

mL polycarbonate bottle) was inoculated with 10 µCi of $^{14}C$-Carbon (Sodium bicarbonate, Perkin Elmer NEC086H005MC) and 4 µCi of $^{33}P$-Phosphate ($H_3PO_4$ in dilute hydrochloric acid, Perkin Elmer NEZ080001MC). The bottles were then placed in blue-screen-on-deck incubators representing 75, 54, 36, 19, 10, 2.7, 1, 0.3 and 0.1 % incident PAR (https://outpace.mio.univ-amu.fr/spip.php?article135) and maintained at constant temperature using a continuous circulation of surface seawater. The

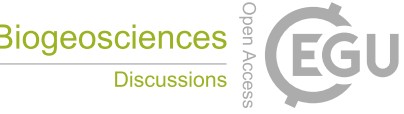



same protocol was used for duplicate 150 mL samples where 150 µL HgCl$_2$ (20 g L$^{-1}$) had been added as a control for non-biological uptake. After 3 to 24 h (the optimal incubation time was determined from a prior time-series experiment), incubations were stopped by the addition of 150 µL of non-radioactive KH$_2$PO$_4$ (10 mmol L$^{-1}$) and dark conditions. Filtrations of 50 mL triplicate subsamples were carried out on 25 mm polycarbonate filters (0.2 µm), placed on DIP-saturated support

GF/F filters, using a low-vacuum pressure < 0.2 bars. Filters were not washed with filtered seawater at the end of the filtration, but pressure was briefly increased to 0.6 bars, to remove non-cellular $^{33}$P radioactivity from the filter. Filters were then placed in low-potassium 6 mL glass scintillation vials (Wheaton) with 500 µL of 0.5 M HCl for 12 hours in order to drive off any unincorporated $^{14}$C. Then, 6 mL of scintillation liquid (Ultima gold MV, Packard) was added and the radioactivity of the filters measured using a scintillation counter Packard Tri-Carb® 2100TR on-board (first count). Initial radioactivity was also

measured on 5 replicates for each profile. Samples were then stored until the second count in the laboratory after $^{33}$P emission became not measurable (12 months). DIC uptake and DIP turnover time were then deduced from the following equations (details in Thingstad et al., 1993; Moutin et al., 2002): T$_{DIP}$ = -Ti/(ln(1-(dpm$^{33}$P-dpm$_{b33P}$)/dpm$_{t33P}$), where T$_{DIP}$ is DIP turnover time (in days), Ti is the incubation time, dpm$^{33}$P is the dpm attributable to the $^{33}$P activity, dpm$_{b33P}$ is the dpm attributable to the blank and dpm$_{t33P}$ is the initial (total) activity of $^{33}$P. V$_{DIC}$= [(dpm$^{14}$C-dpm$_{b14C}$)/dpm$_{t14C}$] * [DIC] / Ti where: V$_{DIC}$ is the C

uptake rate (nmol L$^{-1}$ h$^{-1}$), dpm$^{14}$C is the dpm attributable to the $^{14}$C activity of the filtered sample, dpm$_{b14C}$ is the dpm attributable to the blank, dpm$_{t14C}$ is the initial (total) activity of $^{14}$C added to the sample, [DIC] is the dissolved inorganic carbon concentration of the sample, and Ti is the incubation time. The daily surface photosynthetic available radiation (SPAR) data were used to estimate the daily primary production (PP) values from the PP rates obtained with short time incubation durations using a conversion model (Moutin et al., 1999).

*N$_2$ fixation rates*

N$_2$ fixation rates were measured using the $^{15}$N$_2$ tracer method (Montoya et al., 1996) adapted and precisely described in Bonnet et al. (this issue). Rapidly, seawater was collected in triplicates from the Niskin bottles in 2.3 L polycarbonates bottles at 6 depths (75 %, 54 %, 19 %, 10 %, 1 %, and 0.1 % surface irradiance levels), like for PP measurements. 2.5 mL of $^{15}$N$_2$ gas (99

atom% $^{15}$N, Eurisotop) were injected in each bottle through the septum cap using a gas-tight syringe. All bottles were shaken 20 times to facilitate the $^{15}$N$_2$ dissolution and incubated for 24 h from sunrise to sunrise. To avoid any possible rate underestimation due to equilibration of the $^{15}$N$_2$ gas with surrounding seawater, final ™$^{15}$N enrichment in the N$_2$ pool was quantified for each profile in triplicates at 5 m and at the deep chl a maximum (DCM). After incubation, 12 mL of each 4.5L bottle were subsampled in Exetainers, fixed with HgCl$_2$ and stored upside down at 4°C in the dark and analyzed onshore within

6 months after the cruise according to Kana et al. (1994) using an Membrane Inlet Mass Spectrometer. Incubation was stopped by gentle filtration of the samples onto pre-combusted (450 °C, 4 h) Whatman GF/F filters (25 mm diameter, 0.7 µm nominal porosity). Filters were stored in pre-combusted glass tubes at -20 °C during the cruise, then dried at 60 °C for 24 h before analysis onshore by EA-IRMS on an Integra2 (Sercon Ltd). The detection limit associated with the measurement was 0.14 nmol L$^{-1}$ d$^{-1}$. The accuracy of the EA-IRMS system was systematically controlled using International Atomic Energy Agency

(IAEA) reference materials, AIEA-N-1 and IAEA-310A. In addition, the natural δ$^{15}$N of particulate organic N needed for N$_2$ fixation rate calculations was measured on each profile at two depths (surface and DCM).

**3 Results**

**3.1 General annual trends of MLD, SST and SSchl a for the 3 selected areas**

MLD against month in the climatology (Fig. 2a, 2d, 2g) varied annually from around 70 m depth in July during the austral

winter to between 20-40 m during the austral summer for the 3 areas. The OUTPACE cruise from 18 Feb. to 3 Apr. (red lines) sampled during the stratified period characterized by minimal MLD and maximal SST (Fig. 2b, 2e, 2h). SST varied from 24.2





± 0.2 to 28.8 ± 0.3 °C, 23.8 ± 0.5 to 28.3 ± 0.7 °C, 25.9 ± 0.4 to 29.0 ± 0.4 °C between July 2014 and July 2015 for WMA, EMA and WGY, respectively. Mean March 2015, SST of 28.8 ± 0.3 °C, 28.3 ± 0.7 °C and 29.1 ± 0.4 °C are close to the mean conservative temperature measurements measured in the MLD during the OUTPACE cruise of 28.9 ± 0.3 °C, 29.3 ± 0.3 °C and 29.5 ± 0.4 °C for WMA, EMA and WGY, respectively. The mean conservative temperature measurements at 70 m depth

were 25.3 ± 0.3 °C, 24.8 ± 0.9 °C, 26.1 ± 0.9 °C for WMA, EMA and WGY, respectively (Fig. 3a). These values are comparable with the SST measured during the deeper winter mixing in July 2014 of 24.9 ± 0.2, 24.2 ± 0.7 and 26.5 ± 0.2 for WMA, EMA and WGY, respectively (Table 5). Our hypothesis to consider limited exchanges allowing properties to be conservative at 70 m depth seems reasonable for temperature. Expected seasonal upper surface temperature variations calculated from the differences between temperature at the surface and at 70 m depth of 3.6 ± 0.6, 4.5 ± 1.2 and 3.4 ± 1.3 °C for WMA, EMA and

WGY, respectively, agreed relatively well with SST variations between July 2014 and March 2015 of 3.9 ± 0.5, 4.2 ± 1.4 and 2.6 ± 0.6 °C observed (Fig. 2b, 2e, 2h). Following a similar reasoning, we established a relatively good comparison between chl $\underline{a}$ measured at 70 m depth during OUTPACE of 0.217 ± 0.092, 0.091 ± 0.012 and 0.046 ± 0.010 mg m$^{-3}$ for WMA, EMA and WGY, respectively (Fig. 3f), and SSchl $\underline{a}$ obtained during the deeper mixing of 0.173 ± 0.005, 0.121 ± 0.023 and 0.042 ± 0.002 mg m$^{-3}$ for WMA, EMA and WGY, respectively (Fig. 2c, 2f, 2i). SSchl $\underline{a}$ reflected well the expected oligotrophic

gradient with higher values obtained at WMA, lower values at WGY and intermediate values at EMA. The increase of SSchl $\underline{a}$ observed in July seems related to the deep mixing during winter. The comparison between measurements at 70 m depth and in the upper mixed layer during OUTPACE may be used as a first approach to characterize surface seasonal changes of physical and biogeochemical properties of upper surface waters (section 4.3).

### 3.2 General hydrological and biogeochemical conditions allowing for characterization of oligotrophic states of the

**different upper water masses sampled during OUTPACE**

The general hydrological and biogeochemical conditions during OUTPACE provide the means to characterize the oligotrophic states of the different water masses sampled (Table 1). The shallow austral summer MLD varied between 11 and 34 m with a mean of 16.7 m (SD = 6.4 m). The low variation is in agreement with the relatively similar weather conditions and SST along the zonal transect near 20° S (Moutin et al., 2017). The euphotic zone depth (EZD) and the DCM depth (DCMD) deepen from

west to east, from around 70 m to largely deeper than 100 m, indicating the higher oligotrophy of the SP gyre water compared to the MA water. The DCM concentration decreases from west to east but only slightly, from a maximum of 0.40 to a minimum of 0.25 mg m$^{-3}$. A better indicator of oligotrophic conditions is the depth of the nitracline ($D_{NO3}$) which varied between 46 and 141 m, typical of oligotrophic to ultraoligotrophic areas of the world ocean (Moutin et al., 2012, their Fig. 9). A relative homogeneity of the slopes ($S_{NO3} = 47.0 ± 11.5$ µmol m$^{-4}$) was observed (Table 1). Phosphaclines and nitraclines did not match,

as shown by the lower $D_{PO4}$ observed everywhere. No phosphaclines linked with upper water biological processes were determined in the SP gyre because phosphate concentrations above the QL were measured up to the surface. $S_{PO4}$, when measurable, was 2.8 ± 1.0 µmol m$^{-4}$ (Table 1).

The same characteristics are presented for the 3 areas considered (WMA, EMA and WGY) in Table 2 by their means and SD. The DCMD (about 10-20 m below the EZD in all areas) increased from 78 ± 10 m in the WMA to 134 ± 14 m in the WGY,

with an intermediate value of 104 ± 15 m in EMA. $D_{NO3}$ follows the same pattern, with values of 76 ± 10 m, 100 ± 18 m, and 116 ± 18 m, respectively, showing a clear relationship between DCMD and $D_{NO3}$ (Table 2).

The 3 areas considered are characterized by similar trends of conservative temperature, absolute salinity and potential density vs depth between 0-200 m (Fig. 3a, 3b, 3c), i.e an homogeneity in the mixed layer followed by a drastic change at the basis of the mixed layer and a break in slopes around 70 m depth. Temperature increased from the deeper layer to the surface where

higher temperature characterized the austral summer heating, while lower salinity above 70 m depth indicate significant fresh water input from precipitation. The deepening of the DCMD from WMA (dark green) to WGY (blue) with an intermediate value for EMA (light green) demonstrates the westward-eastward gradient of increased oligotrophy (Fig. 3f), reflected as well



as by corresponding $D_{NO3}$ ($D_{NO3} = D_{DIN}$, see section 2.2) at similar depths (Fig. 5b). 0-70 m integrated chl $\underline{a}$ decreased largely from west to east along the transect, from $7.2 \pm 2.1$ mg m$^{-2}$ for WMA to $2.0 \pm 0.6$ mg m$^{-2}$ for WGY, with an intermediate value of $4.6 \pm 0.7$ mg m$^{-2}$ for EMA (Table 2). When integrated over the top 200 m, no difference between chl $\underline{a}$ stocks were noticeable with a mean value for the whole dataset of $19.9 \pm 2.4$ mg m$^{-2}$.

AOU showed similar patterns in all areas with a slight decrease from the surface to a minimum between 50-70 m and an increase below 70 m (Fig. 3e). The values close to zero for the first depths indicated saturation or a light super-saturation following classical rapid exchanges with atmospheric oxygen. The AOU values below, and up to, 70 m at both WMA and EMA, and to 100 m depth at WGY, indicated oversaturation. Between 70 and 200 m, almost linear relationships between AOU and depth were observed for all areas.

## 3.3 C, N, P pools

The dissolved inorganic (upper), dissolved organic (middle) and particulate organic (below) C, N and P (left to right) pools are represented in Fig. 5. For N and P graphs, a Redfield ratio (RR) of 16:1 was systematically applied between N and P axes, allowing for a more direct comparison. DIC in µmol kg$^{-1}$ (Fig. 4a), nDIC (normalized DIC) in µmol kg$^{-1}$ (Fig. 4c) and in µmol L$^{-1}$ (Fig. 5a) showed linear increasing trends with depth in all areas between 70 and 200 m. The specific variations of nDIC

close to the surface will be discussed later. Total alkalinity increased rapidly with depth between 0 and 70 m and was more or less constant below until 200 m (Fig. 4b). Normalized total alkalinity indicated no change in concentration with depth (Fig. 4d), showing that total alkalinity variations were related to fresh water input. Surface $p_{CO2}^{oc}$ was everywhere close or below the average atmospheric $p_{CO2}$ of 383 µatm (Table 5). Nitrate (DIN) was under the QL everywhere in the upper surface until 70 m (Fig. 5b). Then the increase with depth (nitracline) was almost the same in each area (similar slopes, $S_{NO3}$) but did not begin

at the same depth ($D_{NO3}$) as was previously described. Phosphate (DIP) concentrations were largely higher than nitrate concentrations (considering RR) everywhere except close to the surface at WMA and EMA where they reached QL. High DIP concentration around 0.2 µmol L$^{-1}$ in the upper 70 m were observed at WGY (Fig. 5c). The depletion in DIP was higher in EMA than in WMA (Fig. 5c). DOC, DON and DOP concentrations were higher close to the surface (Fig. 5d, 5e, 5f) and decreased almost linearly with depth until 200 m with only slight differences between the different areas, particularly for the

deeper depth measurements where ~50, 4, and 0.07 µmol L$^{-1}$ of DOC, DON and DOP were measured, respectively. The concentration increases in the surface compared to the values at 200 m depth corresponded roughly to around 25, 1.5, and 0.1 µmol L$^{-1}$ of DOC, DON and DOP, respectively (in similar proportions to the RR for N and P, but more than 2-fold higher for C). The particulate organic C, N, and P pools showed similar patterns with depth between 70 and 200 m, but diverged in the upper layer between the different areas (Fig. 5g, 5h, 5i). No to little changes were observed at WGY while significant increases

in concentration close to the surface were observed both in WMA and EMA. The increases in surface water concentrations compared to the value at 200 m depth corresponded roughly to changes around 5, 0.5 and 0.03 µmol L$^{-1}$ of POC, PON and POP, respectively (in relative similar proportions to the RR for C, N and P).

The 0-70 m depth inventories are presented in Table 3. Interestingly, there were really similar C stocks in the 3 areas, both for the dissolved inorganic and dissolved organic pools. The particulate organic C pool was 2 times lower in WGY than in the

MA. Very similar observations are obtained for all N pools. Nevertheless, DIN stocks were negligible in all areas. DIP stocks were different, and higher in the gyre. The other P pools follow the same pattern as C and N pools, i.e. almost identical in the 3 areas concerning the dissolved organic pool and 2 times lower in the gyre for the particulate pool.

## 3.4 C, N, P fluxes

Some major fluxes, PP and N$_2$ fixation rates together with DIP turnover times, are shown Fig. 6. All rates are largely higher

for WMA and EMA than for WGY, where values indicated only slight differences with depth. Conversely, higher PP (Fig. 6a) and N$_2$ fixation (Fig. 6b) rates were measured close to the surface and rapidly decreased with depth reaching negligible values



below 50 m and beyond for WMA and EMA. $T_{DIP}$ values of around 100 days for WGY contrast with lower values for WMA and EMA upper waters close to or even below 2 days (Fig. 6c).

Particulate matter and swimmer mass fluxes collected with sediment traps are presented in Table 4 with C, N, and P partitioning. Large variability exists between measurements as shown by the minimum and maximum values obtained.

Nevertheless, a mean particulate matter mass flux of 48 mg d$^{-1}$, three times higher in the MA compared to WGY, was obtained, in good agreement with the higher PPrates and biomass in the MA compared to the gyre. Swimmer mass fluxes were also highly variable and represent, as a mean, 9.7 (min: 0.7, max: 26.0) times more mass (dry weight) per day than the settling particles in the MA, and 4.4 (min: 1.4, max: 7.1) times for WGY. The mean proportion of C, N, and P in the settling organic matter of 106/12.7/1.2 for MA and 106/16.6/0.5 for WGY are in relatively good agreement with the theoretical 106/16/1 RR.

Note that it is also the case for C, N, and P proportions in swimmers both for MA (106/15.8/0.7) and WGY (106/19.9/0.7), particularly when P measured in the supernatant was added to the swimmer (see * in Table 4). Otherwise, very low and improbable P contents were found in the swimmers.

## 4 Discussion

### 4.1 A significant biological carbon pump in the WTSP fueled by $N_2$ fixation

We use the surface $p_{CO2}^{oc}$ expected seasonal changes between austral winter and summer in order to draw a first picture of the role of the biological pump in the WTSP. Surface $p_{CO2}^{oc}$ is determined by temperature and salinity changes, and by processes affecting the DIC and alkalinity concentrations, which includes gas exchange, the biological pump, lateral and vertical advection, and mixing (Sarmiento and Grüber, 2006). We will consider that the horizontal spatial scale is large enough to avoid considering lateral advection. Numerical horizontal particle experiments integrating several months of satellite data

using Ariane (Rousselet et al., this issue) together with the relative homogeneity of SST along the 4000 km water transect (Moutin et al., 2017) provides support for this first assumption. Furthermore, we will consider that the influence of salinity changes on the "soft tissue" pump is negligible as generally considered (Sarmiento and Grüber, 2006).

Upper surface temperature variations between the 2014 austral winter and the 2015 austral summer period were estimated to

be 3.6 ± 0.6, 4.5 ± 1.2 and 3.4 ± 1.3 °C for WMA, EMA and WGY, respectively. Estimated winter $p_{CO2}^{oc}$ were 372, 355 and 364 µatm (Table 5). Following Takahashi (1993) calculation ($\Delta p_{CO2}^{oc}{}_{|Thermal} \approx p_{CO2}^{oc} * 0.0423 * \Delta T$) considering a closed system with constant DIC and Alk, we estimate an increase in $p_{CO2}^{oc}$ of +57, +68 and +52 µatm following summer warming for WMA, EMA and WGY, respectively. The seasonal warming should result in an ~60 µatm increase of $p_{CO2}^{oc}$ which is not observed for any group of stations, indeed the differences in $p_{CO2}^{oc}$ were of 366 – 372 = -6, 376 – 355 = +21 and 390 – 364 = +26 µatm

between winter and summer for WMA, EMA and WGY, respectively (Table 5). The differences were obtained from normalized DIC and Alk measured during the OUTPACE cruise in the MLD, and estimated from the expected normalized winter DIC and Alk. The lower than expected $p_{CO2}^{oc}$ changes suggest that the seasonal variations of $p_{CO2}^{oc}$ due to SST changes are counterbalanced by a seasonal reduction due to DIC and/or Alk changes. We can estimate this term by removing $p_{CO2}$ changes due to thermal variation from the observations ($\Delta p_{CO2}^{oc}{}_{|DIC,Alk} = \Delta p_{CO2}^{oc}{}_{|observed} - \Delta p_{CO2}^{oc}{}_{|thermal}$), resulting in -63, -47 and

-26 µatm for WMA, EMA and WGY, respectively. The negative signs imply a decrease in DIC or an increase in Alk between winter and summer. When normalized, we do not observe any difference in Alk with depth (Fig. 4d), suggesting that seasonal salinity changes due to large precipitation may explain the small change in Alk observed (Fig. 4b). Therefore, the carbonate pump does not seem to play a significant role in the WTSP and consequently, we expect a major role of the "soft tissue" pump and thus DIC variations. Considering a Revelle factor $\gamma_{DIC}$ of 9.5, we calculate DIC changes of -35.8, -28.0 and -15.0 µmol kg$^{-}$

$^{1}$ ($\Delta DIC = DIC/p_{CO2}^{oc} \gamma_{DIC} * \Delta p_{CO2}^{oc}{}_{|DIC,Alk}$) necessary to explain the changes in $p_{CO2}^{oc}$ observed. We observe indeed a decrease in nDIC concentration of 32.9, 25.7 and 15.3 µmol kg$^{-1}$ (Table 5) for WMA, EMA and WGY, respectively (37.0, 30.0 and



18.7 µmol L$^{-1}$, Table 7, Fig. 5a) between the estimated winter concentration and the mean value measured during the OUTPACE cruise that may explain the negative sign, and the order of magnitude of the DIC changes. This result based on estimated winter values is reinforced by the fact that winter DIC from NDP-094 climatology of 2006.4 ± 0.7, 2000.9 ± 3.0 and 2004.7 ± 9.9 µmol kg$^{-1}$, are really close to our estimates for winter conditions, 2007.5 ± 3.0; 2009.6 ± 9.6 and 2008.9 ± 3.7

µmol kg$^{-1}$, for WMA, EMA and WGY, respectively (Table 5). TA also showed good agreement, 2335.4 ± 0.2, 2333.6 ± 1.7 and 2343.4 ± 8.6 µmol kg$^{-1}$ from NDP-094 climatology, and 2332.4 ± 5.0, 2344.1 ± 6.5 and 2350.8 ± 2.7 µmol kg$^{-1}$ with our estimates for winter conditions. The differences between climatological $p_{CO2}^{oc}$ and our estimates for winter conditions are higher (Table 5) and can be related to differences in temperature (SST from NDP-094 climatology, SST from MODIS Aqua, T from our estimates). If $p_{CO2}^{oc}$ are calculated from DIC and Alk (NDP-094 climatology) with SST from MODIS Aqua (361,

344 and 371 µatm) or our estimated temperatures (366, 353 and 368 µatm), the values are really close to our estimated winter upper surface $p_{CO2}^{oc}$ (372, 355 and 364 µatm for WMA, EMA and WGY, respectively) (Table 5). Upper surface estimated DIC seasonal changes may explain why counterintuitive low seasonal $p_{CO2}^{oc}$ changes were obtained despite significant increases in temperature. What is therefore controlling the decrease in nDIC? Is it gas exchange at the air-sea interface, mixing, and/or the biological pump?

Gas exchange may be excluded because surface water $p_{CO2}^{oc}$ ranged 355-390 µatm while the $p_{CO2}^{atm}$ is 383 µatm with almost no seasonal variations (Table 5). Therefore, surface waters are close to saturation at WGY or under-saturated in the MA all year and will uptake $CO_2$ from the atmosphere, and as a result DIC should then increase, which is not observed. Thus, our observations are more biological in origin, but we have an inconsistency. The significant decrease in nDIC (Fig. 5a and Table 7), indicating a significant biological soft tissue pump, coincided with no significant changes in nitrate concentration, which

were ≤0.03 µmol L$^{-1}$ in all areas (Fig. 5b, Table 7) indicating no or almost no nitrate input by deep winter mixing. Considering the low nitrogen input by upward nitrate turbulent diffusion (see later), we have to consider another nitrogen source, $N_2$ fixation (Fig. 6b), which is really high in the upper water of the WTSP, recently identified as a hot spot for $N_2$ fixation (Bonnet et al., 2017).

The estimated seasonal nDIC (ΔDIC) variations for the MA waters of 32.9 and 25.7 µmol kg$^{-1}$ for WMA and EMA,

respectively, can be compared to those measured in oceanic gyre time-series sites. They are higher than the ΔDIC~15 µmol kg$^{-1}$ observed at the HOT station in the North Pacific subtropical gyre near Hawaii (Dore et al., 2003) and close to the ΔDIC~30 µmol kg$^{-1}$ observed at BATS in the subtropical North Atlantic gyre near Bermuda (Bates et al., 2012), where ΔDIC is at least partially attributable to nitrate from below (Sarmiento and Grüber, 2006). Interestingly, the estimated amplitude of surface DIC seasonal change for the MA is only 2 times lower than the around 50 µmol kg$^{-1}$ DIC decrease measured between March

and April in the northern Atlantic (Merlivat et al., 2009), in an area known to experience a large bloom of phytoplankton. The biological "soft tissue" carbon pump, fueled almost exclusively by $N_2$ fixation (see section 4.2), therefore plays a significant role in the WTSP.

## 4.2 A net sink of atmospheric $CO_2$ mainly driven by zooplankton migration in the MA

Quantification of the major biogeochemical fluxes on a daily basis allows for the establishment of some conclusions concerning

the upper biogeochemical cycles of C, N, and P (Table 6). C-budgets of the 0-70 m upper layer showed that the MA area appears as a net sink of atmospheric $CO_2$. Atmospheric carbon input in the ocean was the major flux in the WMA. Sediment trap POC export was one order of magnitude higher than POC or DOC export by turbulent diffusion, which represented only 7-12 % of the total organic export. Without considering any additional flux, the budget resulted in a surprisingly daily net accumulation of carbon of 0.9 mmol m$^{-2}$ d$^{-1}$ for WMA, and a quasi-equilibrium for EMA and WGY. Note that the accumulation

at WMA resulting in an increase of only several nmol L$^{-1}$ d$^{-1}$, is largely below what we are able to measure at the present time, and longer time scale are thus needed to observe and study the changes (section 4.3). Else we need to explain the estimated



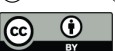

net seasonal decreases of DIC in the upper surface waters and of the total carbon pool between all areas (section 4.1, Table 7)) which implies that we probably missed an export flux, particularly for WMA.

Averaged integrated PP (IPP) rates were $33.3 \pm 12.1$, $26.4 \pm 16.2$ and $6.5 \pm 2.4$ mmol m$^{-2}$ d$^{-1}$ and export by settling and turbulent diffusion (Table 6, in µmol m$^{-2}$ d$^{-1}$) represented only 1.2, 1.2 or 0.3 mmol m$^{-2}$ d$^{-1}$, for WMA, EMA and WGY, respectively.

The organic matter exported daily compared to IPP represented 3.6, 4.5 and 4.6 %, respectively, in good agreement with previous measurements in oligotrophic areas (Moutin and Raimbault, 2002, Karl et al., 2012) with a high proportion relative to particles settling, 3.3, 4.1 and 4.1 %, rather than turbulent diffusion. Furthermore, no large increase in phytoplankton biomass (chl a) was observed during the whole year in the upper surface (Fig. 2c, 2f, 2i). Chl a varied only between 0.05 and 0.20 mg m$^{-3}$ in the MA, suggesting a strong top-down control by zooplankton able to maintain pigment concentration in a quasi-steady

state for many months (Banse et al., 2012). Therefore, as has already been observed in the equatorial Pacific (Landry et al., 2011), it is not unconceivable to consider an equilibrium between phytoplankton production and grazing by mesozooplankton. Both sediment trap data (Table 4, last column) and ADCP measurements (not shown) indeed indicate large zooplankton diel vertical migration, the latter being widespread in the ocean and forming a fundamental component of the biological pump generally overlooked in global models (Bianchi et al., 2013). The ADCP data clearly shows vertical migration from the upper

level depths down to around 500 m when light increases at the 2 stations LD A & LD C, with the reverse migration back to the upper levels when light decreases. The objective for mesozooplankton is to feed at night in order to avoid predators. Additionally, while mesozooplankton spend half of the time at around 500 m depth, they respire and lose carbon. Around 25% of their biomass in term of carbon is considered to be lost through respiration each day (Ikeda, 2014; Pagano, pers. com). Considering that half of this loss (12.5 %) happens at 500 m depth following ingestion of the water column's whole PP(new

biomass) in the upper surface, it may explain fluxes of 4.2, 3.3 and 0.8 mmol m$^{-2}$ d$^{-1}$, largely able to significantly influence the daily budgets (Table 6 in µmol m$^{-2}$ d$^{-1}$). The estimated downward flux of carbon from the euphotic zone due to mesozooplankton diel vertical migrators was at least one order of magnitude higher than the 0.6-1.1 mmol C m$^{-2}$ d$^{-1}$ reported for the equatorial Pacific (Zhang and Dam, 1998). But the mean C export by swimmers of 632 % (MA) and 876 % (WGY) relative to the passive flux measured (Table 4) was also largely higher than the 15-30 % reported at ALOHA station (Al-

Mutairi and Landry, 2001). Furthermore, the numerous species of mesozooplankton observed during OUTPACE were not all known to migrate (Carlotti et al., this issue) and temperature dependence on metabolic rates (Ikeda, 2014) needs also to be taken into account (i.e. a slower respiration at depth in colder temperature). Even if considerable uncertainty remains, a predominant role of mesozooplankton in the transfer of carbon (biological pump) is suggested by these data in the WTSP, and particularly in the MA waters.

Except for the WMA area, there were no DIN gradients around 70 m depth and therefore no nitrate input from below by turbulent diffusion (Table 6). Nitrogen input by N$_2$ fixation was by far the largest input of new nitrogen (at least 83%) and reached among the largest values measured everywhere in the open ocean (Bonnet et al., 2017; Caffin et al., this issue, Knapp et al., this issue). A net daily accumulation of nitrogen is estimated for MA and equilibrium for WGY. Zooplankton diel migrations may also play a significant role in daily N budgets through defecation, excretion or mortality in depth (Caffin et

al., this issue; Valdes et al., this issue). Averaged integrated N$_2$ fixation rates were $0.64 \pm 0.21$, $0.45 \pm 0.27$ and $0.04 \pm 0.04$ mmol m$^{-2}$ d$^{-1}$ for WMA, EMA and WGY, respectively. The really high N$_2$ fixation rates in the MA, compared to other areas in the world (Bonnet et al., 2017), may provide the nitrogen required for primary production, creating the necessary decrease in p$_{CO2}^{oc}$ to stimulate CO$_2$ invasion.

The daily P-budgets of the 0-70 m upper layer showed losses greater than inputs, in complete opposition with daily C and N

budgets showing accumulation in the WMA (Table 6). This main observation indicates why this element, compared to carbon and nitrogen, may rapidly become a limiting factor for biological production and specifically of the input of nitrogen by N$_2$ fixation in the MA (Moutin et al., 2008). Nevertheless, the mean particulate P export seemed relatively high (Table 6) and





should be considered with caution considering the huge range of variation, from 0.6 to 68.9 µmol m$^{-2}$ d$^{-1}$, for only 8 measurements in the MA.

**4.3 Estimated seasonal trends of the major biogeochemical stocks and fluxes**

As already written, the dashed lines in all Figs. 3, 4 and 5 indicate the upper surface expected values for all variables during

the 2014 austral winter, and allow for evaluation of the temporal variation toward the 2015 austral summer season (full lines) in each area corresponding to the OUTPACE dataset. The strong hypothesis allowing this first-order estimation of seasonal variation was presented in section 2.1, validated for SST and chl a variations in section 3.1, and shown to give good agreement with upper surface DIC expected seasonal changes (section 4.2).

Conservative temperature (Fig. 3a) increased everywhere, but more for WMA and EMA than for WGY, while absolute salinity

decreased everywhere. Potential density values were similar in each area at 70 m depth. Similar mean depths of convection were estimated for the three areas (min of 68 m at LD A and max of 73 m at LD C), and justified the mean value of 70 m taken into account for the whole OUTPACE area. The rapid exchanges of oxygen between ocean and atmosphere pre-empted significant seasonal changes in the upper surface (Fig. 3d, 3e). The vertical homogeneous chl a concentration expected in winter (Fig. 3f) was shown to be in good agreement with climatological SSchl a (section 3.1). Part of the relatively high chl a

concentration estimated in July 2014, specifically in WMA, is likely linked to enhanced vertical winter mixing from the DCM. The seasonal C, N, and P pool changes may be followed in concentration in Fig. 5 but are easier to discuss as 0-70 m water column inventories (Table 8). As previously indicated, DIC decreased in all areas but more in the west than in the east (Fig. 5a), following the already described oligotrophic gradient clearly shown both in biomass (Fig. 3f) and in PP (Fig. 6a). The DIC decrease was partially compensated by the increase in organic concentrations, with the increase of the dissolved

concentrations (Fig. 5d) being larger than the particulates (Fig. 5g). No increase in the particulate carbon concentration was found for WGY. The decrease of TC (representing the sum of all pools) between winter and summer indicated that 68.1, 61.9 and 68.3 % of ΔDIC were lost from the upper layer, i.e only 31.9, 38.1 and 31.7 % accumulated in the organic C pools for WMA, EMA and WGY, respectively (Table 8). Therefore, organic matter accumulation may partly explain why large input of atmospheric carbon did not result in DIC accumulation in the MA waters. It may partly explain why the total carbon pool

decreased so much seasonally. Following the RR, DIN decreases of 236, 198 and 109 mmol m$^{-2}$ might be expected from the DIC decreases. Indeed, the DIN decreases were around 0-2 mmol m$^{-2}$, which is in concordance with very poor DIN replenishment of the upper water column. Conversely, increases of the PON stocks on the same order of magnitude as the RR predicts from POC stocks for WMA and EMA were observed (12.0 and 7.3, compared to RR = 6.6), with a small PON decrease for WGY. The largest increases for the organic pools were for the dissolved phase in all areas (Table 8). DOC accumulation

was 3.8 and 8.1 times higher than POC accumulation for WMA and EMA, respectively. Only DOC accumulated at WGY, but with a change two times lower in magnitude than in the MA waters (Table 8). A relative stronger dissolved organic carbon production compared to particulate production may be reached in oligotrophic areas, depending largely on light and nutrient availabilities (Carlson, 2002). In oligotrophic areas, characterized by low export of particulate organic matter, relatively large dissolved organic matter production, and heterotrophic bacteria often limited by nutrients (Van Wambeke et al., 2002), DOC

may accumulate (Copin-Montégut and Avril, 1993; Marañón et al., 2005, Pujo-Pay et al., 2011), which is indeed observed (Fig. 5d). Dissolved organic carbon accumulation reached 391, 445 and 220 mmol m$^{-2}$ over 8 months (Table 8) which dispersed over 70 m gives a mean 8-month accumulation of 7.0, 7.1 and 3.0 µmol L$^{-1}$ for the 0-70 m water column. These values, while lower, are of the same order of magnitude of DOC concentration changes observed in the upper mixed layers of 10.1, 9.3 and 5.0 µmol L$^{-1}$ for WMA, EMA and WGY, respectively (Fig. 5d, Table 7). Interestingly, the western SP was recently shown as

a localized refractory dissolved organic carbon sink (Hansell and Carlson, 2013).

No significant DIN inventory changes were observed while large increases in the DON stocks and similar but relatively lower increases were observed for the PON stocks for WMA and EMA (Table 8, Fig. 5e, 5h). The TN evolution was a net increase



of inventories between winter and summer, of 49 and 34 mmol m$^{-2}$ for WMA and EMA, respectively. No significant changes of the N pools were observed at WGY (Table 8, Fig. 5b, 5e, 5h). A decrease of DIP stocks was observed in the MA waters between the winter and summer, with no significant change for WGY (Table 8). Following the RR, DIP decreases of 14.7, 12.3 and 6.8 mmol m$^{-2}$ might be expected from the DIC decreases. Indeed, the DIP decreases were less, 5.9 and 3.1 mmol m$^{-2}$ for WMA and EMA, and no decrease observed at WGY. The DIC decreases are probably only partially related to the DIP decreases in the MA. As for C and N, the largest organic P inventory increases were for the dissolved phase (Fig. 5f, Table 8). Nevertheless, the changes were close to the SD calculated for the mean concentrations and should be considered with caution. As an example, the 1.8 mmol m$^{-2}$ increase in DOP concentrations for EMA (Table 8) corresponds to the difference between 11.6 ± 1.1 mmol m$^{-2}$ during winter and 9.8 ± 2.0 mmol m$^{-2}$ during summer. Note that the SD reported is the maximum SD calculated at each season (Table 8). Small or no decreases in the organic P pools were observed for WGY. Finally, it is clear that seasonal C losses were not compensated by organic carbon accumulation in the 0-70 m layer. Therefore, organic carbon production, which represents by far the largest flux in each area, should be linked with an efficient export from the upper layer, not directly related to RR.

We now try to connect the seasonal variations of C, N, and P stocks with the estimated C, N, and P fluxes in order to draw first-order budgets and characterize the main seasonal trends in the WTSP. Our very simple model considers an instantaneous winter mixing followed by 8 months (240 days) of C, N, and P fluxes at the same rates as the mean rates measured during the OUTPACE cruise. All fluxes expressed in mmol m$^{-2}$ and corresponding to the 8-month period defined (July 2014-March 2015) are synthesized in Fig. 7. Accumulation rates are presented inside the boxes and input and output fluxes outside the boxes with arrows for direction (+ for input, - for output). The X value corresponds to the flux necessary to reach equilibrium in each box.

The main question is still how can we explain the large DIC losses in all areas whereas we got a significant DIC input by winter convection and turbulent diffusion, low export of organic matter by settling or turbulent diffusion, and a $p_{CO_2}^{oc}$ lower than or equal to the $p_{CO_2}^{atm}$ meaning a DIC enrichment by atmospheric exchanges, and furthermore no significant input of DIN from below in the 0-70 m upper layer?

The source of new N required to sustain new PP is clearly N$_2$ fixation (Fig. 7b, 7e, 7h). Converted in C using the RR of 6.6, new production may represent 12.8, 11.3 and 4.2 % of IPP of 7.94, 6.34 and 1.56 mol m$^{-2}$ for the 8-month period in the WMA, EMA and WGY, respectively. A new production ≤ 5% is typical of strong oligotrophic conditions (Moutin and Raimbault, 2002) while above 5% is related to more productive areas or areas with high N$_2$ fixation rates (Karl et al., 2012). Taking into account the fact that the previous values are for 8 months only, we can estimate annual productions of 145, 116 and 28 gC m$^{-2}$ y$^{-1}$ for WMA, EMA and WGY, respectively, close to the average rate of 170 gC m$^{-2}$ y$^{-1}$ reported for the ALOHA station in the North Pacific central gyre (Karl et al., 1996) and to the 86-232 gC m$^{-2}$ y$^{-1}$ range reported for the Mediterranean Sea at the DYFAMED site (Marty and Chiavérini, 2002), known as oligotrophic areas.

Having found the source of new N, in order to answer the question regarding DIC losses, a first hypothesis may be to consider episodic high export of matter, in complete contradiction with our initial postulate. We cannot completely discard this hypothesis specifically because no seasonal data are available at the present time and also because episodic yet large export fluxes have already been reported in other oligotrophic areas (Böttjer et al., 2017). Nevertheless, the relative constant chl a concentration during the entire period considered in the upper water column (Fig. 2c, 2f, 2i), where most of the production is likely to occur (Fig. 6a), preferentially suggests relatively constant production and therefore export. Furthermore, the C, N, and P proportions of the X fluxes (Fig. 7) in all areas are completely different from RR, even in an opposite sense for P (Fig. 7c, 7f, 7i), suggesting that such C fluxes were not directly related to organic matter settling.

A second hypothesis considers a major role of zooplankton migration in the transfer of carbon. It seems that it is the only way to explain significant C losses with proportionally lower N losses and no P losses (Fig. 7). If, as already suggested, a quasi-steady state between phytoplankton and zooplankton productions is considered, which means that IPP is totally grazed by zooplankton, and that 12.5% of the carbon was lost by zooplankton respiration during its stay in depth, then we found C losses




of 993, 793 and 195 mmol m$^{-2}$ for that period. These numbers are of the same magnitude order than the X values of 1274, 821 and 426 mmol m$^{-2}$ determined from seasonal C budgets (Fig. 7a, 7d, 7g). Because the C lost in that case, by respiration, is independent from N or P losses, it may explain the discrepancies observed between the C, N and P fluxes. This, together with the observed zooplankton migration through ADCP data, definitely suggests that zooplankton may be a preponderant actor in

the transfer of carbon from the upper layer to the interior of the ocean in the WTSP.

**4.4 Iron and phosphate availabilities as key factors controlling the N input by N$_2$ fixation and the biological carbon pump in the WTSP**

The western SP is known as an iron rich area (Wells et al., 1999). Iron concentrations measured during the DIAPALIS cruises near New Caledonia (M. Rodier unpubl. data in Van den Broeck et al., 2004) were higher than concentrations reported in the

sub-tropical North Pacific (Landing and Bruland, 1987). Average iron concentrations of 0.57 nmol L$^{-1}$ were reported in the upper surface waters of the WTSP (Campbell et al., 2005), higher than the ~0.1 nmol L$^{-1}$ measured in the upper 350 m water column of the SP gyre (Blain et al., 2008). The Equatorial Undercurrent, which originates near Papua New Guinea, close to New Caledonia, is known to be a source of iron in the SP Ocean (Wells et al., 1999; Ganachaud et al., 2017). Nevertheless, atmospheric deposition fluxes of iron are very low (Duce and Tindale 1991, Wagener et al., 2008). During OUTPACE, the

apparent contradiction between low atmospheric deposition of iron and high surface water iron concentration was solved. The high iron average concentration within the photic layer in the MA (1.7 nmol L$^{-1}$) compared to WGY (0.3 nmol L$^{-1}$) was shown to be related to an influence of hydrothermal sources at shallower depths than commonly associated with volcanic activities (Guieu et al., in revision) confirming the importance of hydrothermal contribution to the oceanic iron inventory (Tagliabue et al., 2010; Tagliabue et al., 2017). Iron is a major component of the nitrogenase enzyme that catalyzes N$_2$ fixation (Raven,

1988). The high iron concentration likely alleviates the iron limitation of N$_2$ fixation in the WTSP, again considered as a hot spot of N$_2$ fixation (Bonnet et al., 2017).

Phosphate turnover time (T$_{DIP}$) represents the ratio between natural concentration and uptake by planktonic species (Thingstad et al., 1993) and is considered the most reliable measurement of phosphate availability in the upper ocean waters (Moutin et al., 2008). Phosphate availability in the MA, characterized by DIP < 50 nmol L$^{-1}$ and T$_{DIP}$ reaching below 2 days, is largely

lower than in the SP gyre with DIP concentration above 100 nmol L$^{-1}$ and T$_{DIP}$ in the order of magnitude of months (Fig. 6c) as already reported (Moutin et al., 2008). Phosphate availability, as well as primary production, were shown to follow the same seasonal patterns close to New Caledonia in the MA, suggesting that in this iron-rich area known to sustain high N$_2$ fixation rates, phosphate may appear as a key factor controlling carbon production (Van den Broeck et al., 2004). Indeed, a seasonal pattern of phosphate availability with higher values (Low DIP, High T$_{DIP}$) related to winter mixing and lower value (Higher

DIP, Lower T$_{DIP}$) during the stratified period was suggested to control *Trichodesmium* spp. growth and decay in the SP near New Caledonia (Moutin et al., 2005). A T$_{DIP}$ below 2 days was shown to be critical for *Trichodesmium* spp. growth (Moutin et al., 2005). T$_{DIP}$ below or close to 2 days was measured in the MA upper waters during the OUTPACE cruise (Fig. 6c) and T$_{DIP}$ as low as several hours was measured at LD B station and has been related with the strong biomass and specifically *Trichodesmium* spp. decline observed at this station (de Verneil et al., 2017). With T$_{DIP}$ around or even below 2 days, the MA

appears as a low P area during the stratified period indicating a probable role of phosphate availability in the control of nitrogen input by the nitrogen fixers. The higher iron availability in the MA is probably the main factor allowing N$_2$ fixation to occur, and phosphate availability the main factor controlling its rate. A T$_{DIP}$ of 2 days corresponds to the lowest value reported at ALOHA station in the NP (Table 2 in Moutin et al., 2008) where phosphate availability is considered to play a dominant role in the control of nitrogen fixers (Karl et al., 1997; Karl, 2014). T$_{DIP}$ reached several hours which is closest to the phosphate

availability of the Mediterranean Sea or the Sargasso Sea known for a long time for their phosphate deficiency (Wu et al., 2000; Moutin et al., 2002). While phytoplankton and heterotrophic bacterioplankton may appear N-limited (Van Wambeke et al., this issue; Gimenez et al., this issue), the low availability of phosphate in the upper water of the WTSP during the stratified





period likely controls the biomass of nitrogen fixers and ultimately the input of nitrogen by this process. In a recent mesocosms experiment, large increases in $N_2$ fixation rates, PP rates and carbon export were obtained after a DIP enrichment of WTSP waters (Berthelot et al., 2015). Nevertheless, several days were necessary to measure significant increases indicating that regular short term experiments to establish nutrient limitation as usually operated (Moore et al., 2013), may not be relevant in

WTSP conditions (Gimenez et al., 2016).

The high DIP low DIN (excess P or high P*) content of water was suggested to be a preliminary condition allowing $N_2$ fixation to occur (Redfield, 1934; Capone and Knapp, 2007; Deutsch et al., 2007), and is a characteristic of surface waters of the South Equatorial current flowing from the east to the west in the SP due to intense denitrification related to one of the main OMZ (Oxygen Minimum Zone) area in the East Pacific (Codispoti et al., 2001). The alleviation of iron limitation when waters

originating from the east reach the WTSP was considered as the main factor explaining the hot spot of $N_2$ fixation observed in the OUTPACE area (Bonnet et al., 2017). The strong nitracline and phosphacline depth differences (Table 1), associated with winter mixing down to around 70 m, allows us to estimate a replenishment of DIP on the order of magnitude of ΔDIP (5.9 mmol m$^{-2}$ for WMA and 3.0 mmol m$^{-2}$ for EMA; Fig. 7c, 7f) largely above the vertical input by turbulent diffusion (around 0.7 mmol m$^{-2}$) together with no DIN replenishment. Alone, these DIP fluxes may support $N_2$ fixation of 94.4 and 48.0 mmol

m$^{-2}$ during this period (following RR), on the order of magnitude of the fluxes of 154 and 108 mmol m$^{-2}$ calculated for WMA and EMA (Fig. 7b, 7e), respectively. While horizontal advection of high DIP low DIN waters from the SP gyre toward the iron-rich WTSP was suggested to create the environmental conditions favourable for diazotroph growth (Moutin et al., 2008; Bonnet et al., 2017), we here suggest that local seasonal winter mixing may also play a significant role in providing excess P to the upper waters, and therefore in controlling nitrogen input by $N_2$ fixation and therefore the associated carbon cycle.

Phosphate availability appears, in the high iron MA, as the ultimate control of the biological carbon pump. The simulations of the main C, N, and P fluxes at LD A and LD C using a 1DV model with similar physical forcing strengthen the idea of strong seasonal variations being able to explain the control of $N_2$ fixation and carbon fluxes by the availability of phosphate (Gimenez et al., this issue).

### 4.5 Toward reconciliation between simulations and observations?

During the past 10 years, global biogeochemical model simulations suggested relatively high $N_2$ fixation in the SP gyre and low fixation in the western part of the Pacific Ocean (Deutsch et al., 2007; Grüber 2016) in contradiction with the little data then available. While the decrease in P* toward the centre of the gyre observed during the BIOSOPE cruise (ETSP toward the central gyre 10-30°S in latitude) corresponds to the trend observed by Deutsch et al. (2007), $N_2$ fixation in the simulation, with minimum values found on the edge and maximum values found in the centre of the gyre, was contrary to our observations

(Moutin et al., 2008). The high $N_2$ fixation expected in the ETSP, because "downstream of OMZs, surface waters that initially carry a surplus of phosphorus (because of subsurface denitrification) lose this excess gradually through $N_2$ fixation" (Deutsch et al., 2007), was not confirmed by isotopic budgets (Knapp et al., 2016) suggesting an elusive marine $N_2$ fixation (Grüber, 2016). The discovery of a hot spot of $N_2$ fixation in the whole WTSP covered by the OUTPACE transect and other cruises in the Coral Sea (Bonnet et al., 2017) allow us to consider a larger spatial coupling between denitrification and $N_2$ fixation than

previously thought (Deutsch et al., 2007). Taking into account the role of iron to allow (or not) $N_2$ fixation to occur, seems indispensable for the reconciliation between simulations and observations (Dutkiewicz et al., 2012; Monteiro et al., 2011; Weber and Deutsch, 2014). Indeed, these new modelling efforts have identified the WTSP as a unique region with conditions seemingly favourable for significant $N_2$ fixation fluxes (Knapp et al., this issue). Interestingly, the opposite trends between expected $N_2$ fixation and P* observed during the BIOSOPE cruise and possibly attributed to non-Redfieldian processes

(Moutin et al., 2008) may be rather due to horizontal advection and isopycnal mixing of water masses originating from the WTSP and therefore marked by a strong signature of intense $N_2$ fixation (High N* corresponding to Low P*) (Fumenia et al., this issue), in an opposite sense than the more well-known and studied influence of water masses marked by a strong signature





of intense denitrification originating from the OMZ (Yoshikawa et al., 2015). Furthermore, the deepening of isopycnals from the east to the west SP (Yoshikawa et al., 2015; Fumenia et al., this issue) suggest a deeper (∼200 m) influence of excess P waters from the SEC in the MA, deeper than previously hypothesised (Moutin et al., 2008; Bonnet et al., 2017). Because isopycnal mixing influence is below the maximum mixing depth estimated in the WTSP (∼70 m), the link between N sink in

the east and N source in the west imply longer time scales than the one associated to surface circulation. The N budget of the SP Ocean is of prime interest to understand the efficiency, at the present time, and in the future, of the oceanic biological carbon pump. Getting the budget requires a precise understanding of the general water mass circulation which suffers at the present time from a lack of data, specifically during water mass formation (Fumenia et al., this issue).

### 5 Conclusion

We found a significant biological soft tissue carbon pump in the WTSP despite no winter replenishment of surface waters by DIN. $N_2$ fixation is the major process introducing the necessary N to sustain the biological soft tissue carbon pump allowing oceanic $p_{CO2}^{oc} < p_{CO2}^{atm}$ in the MA and therefore significant atmospheric C input. Thanks to $N_2$ fixation, the WTSP is a significant atmospheric carbon sink. We suggest that zooplankton diel vertical migration at around 500 m depth, and their respiration at depth, may significantly contribute to the transfer of carbon from the upper surface to the ocean interior.

The upper surface waters of the MA sampled during the stratified period were characterized by a DIP availability close to or below the level required for phosphate sufficiency, which contrasts with observations in the central Pacific gyre at the same latitude. We confirmed the geographical trend of limitation of $N_2$ fixation in the SP, from an iron limitation in the east and central SP Ocean, to a P limitation in the west. The limit was clearly shown to be associated with the lower depths of the MA where sufficient iron was provided to upper surface waters to alleviate iron limitation of $N_2$ fixation, probably by hydrothermal

sources at anomalously shallow depths. Extrapolating these data in order to obtain seasonal trends allows us to show that winter vertical mixing, although limited to 70 m depth, may bring sufficient excess P to allow most of $N_2$ fixation to occur. Additionally, more excess P may be locally provided in the upper surface (where $N_2$ fixation was shown to occur predominantly) by winter mixing than by horizontal transport from areas of excess P formation (OMZ). As previously hypothesized (Moutin et al., 2008), the low availability of phosphate in the high iron upper waters of the WTSP during the

stratified period likely controls the biomass of nitrogen fixers and ultimately the input of nitrogen by this process, and the biological pump.

The SP Ocean deserves special attention because of its huge volume of water where the N budget is likely to be controlled by N lost in the east (denitrification) and N gain in the west ($N_2$ fixation). Furthermore, both diazotrophy and denitrification are known to undergo drastic alterations due to climate change. Our data suggest that one better take into account the role of iron

in global biogeochemical models, in order to better reconcile simulations and data, which seems to be the prerequisite to understand at the present time the relationship between N sources and sinks in the SP Ocean. Moreover, it will be of great interest to study future scenarios which consider iron coming from below (hydrothermal sources) rather than from above (atmospheric source) in the WTSP and in the whole SP Ocean. Changes in $N_2$ fixation following changes in dust (iron) supply have been suggested to play a central role in explaining past glacial/interglacial changes in $CO_2$ concentration and earth

temperature. It was considered that $N_2$ fixation on a regional scale would change global nitrogen availability and the biological carbon pump on the time scale of ocean circulation. The direct link between $N_2$ fixation and carbon export through zooplankton diel migration and respiration proposed here for the WTSP, hot spot of $N_2$ fixation, allows for a much closer coupling between $N_2$ fixation and the biological carbon pump, which may in turn require us to consider changes at shorter time scales like the one associated with climate change. The low P availability may appear as the ultimate control of N input by $N_2$ fixation and

therefore on the efficiency of the biological pump in the MA.




**Acknowledgements**

This is a contribution of the OUTPACE (Oligotrophy from Ultra-oligoTrophy PACific Experiment) project (https://outpace.mio.univ-amu.fr/) funded by the French research national agency (ANR-14-CE01-0007-01), the LEFE-CyBER program (CNRS-INSU), the GOPS program (IRD) and the CNES (BC T23, ZBC 4500048836). The OUTPACE cruise

(http://dx.doi.org/10.17600/15000900) was managed by the MIO (OSU Institut Pytheas, AMU) from Marseille (France) and received funding from European FEDER Fund under project 1166-39417. The authors thank Nicolas Metzl for the constructive comments on the manuscript and the SNAPO-CO$_2$ (Service National d'Analyse des paramètres Océaniques du CO$_2$- LOCEAN – Paris). The authors also thank the crew of the R/V L'Atalante for outstanding shipboard operation. G. Rougier and M. Picheral are warmly thanked for their efficient help in CTD rosette management and data processing, as is Catherine Schmechtig for

the LEFE CYBER database management. The satellite-derived data of Sea Surface Temperature, chl a concentration and current have been provided by CLS in the framework of the CNES funding; we warmly thank I.Pujol and G.Taburet for their support in providing these data. Aurelia Lozingot is acknowledged for the administrative work.

We acknowledge NOAA, and in particular R.Lumpkin, for providing the SVP drifter. Argo DOI (http://doi.org/10.17882/42182). All data and metadata are available at the following web address: http://www.obs-

vlfr.fr/proof/php/outpace/outpace.php

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



Table 1. General physical and biogeochemical characteristics of the stations investigated along the OUTPACE transect presented by columns : (1) Short duration (SD) or long duration (LD) station; (2) Station number; (3) CTD-rosette number; (4) MLD: Mixed layer depth (m); (5) EZD: Euphotic zone depth (m); (6) $K_z$: Mean 40-200 m vertical eddy diffusivity (m² d⁻¹); (7) $K_z$ error (m² d⁻¹); (8) DCMD, Deep chl $\underline{a}$ maximum depth; (9) DCMC, Deep chl $\underline{a}$ maximum concentration: chl $\underline{a}$ concentration at the DCMD (mg m⁻³); (10 and 11) Ichl $\underline{a}$, integrated (0-70 m) and (0-200 m) total chl $\underline{a}$ concentration (mg m⁻²); (12-24) $D_x$, $eD_x$, $S_x$, $eS_x$, $N_x$ and $r_x^2$: characteristics of nutriclines (depths in m where $NO_3$ or $PO_4$ reaches zero and slopes in µmol m⁻⁴ with associated errors, N: number of samples for the linear relationship, $r^2$ : correlation coefficient). QL: Quantification limit; nd : no data; nc : not calculated (linear relationship not established); na : nutrient above QL at surface.

| Station | | CTD number | MLD (m) | EZD (m) | $K_z$ (m² s⁻¹) | $eK_z$ (m² s⁻¹) | Chl $\underline{a}$ DCMD (m) | DCMC (mg m⁻³) | I 0-70m (mg m⁻²) | I 0-200m (mg m⁻²) | Nitracline $D_{NO3}$ (m) | $eD_{NO3}$ (m) | $S_{NO3}$ (µmol m⁻⁴) | $eS_{NO3}$ (µmol m⁻⁴) | N | $r^2$ | Phosphacline $D_{PO4}$ (m) | $eD_{PO4}$ (m) | $S_{PO4}$ (µmol m⁻⁴) | $eS_{PO4}$ (µmol m⁻⁴) | N | $r^2$ |
|---|---|---|---|---|---|---|---|---|---|---|---|---|---|---|---|---|---|---|---|---|---|---|
| SD | 1 | out_c_006 | 12 | 70 | 9,37E-06 | 1,10E-05 | 88 | 0,40 | 5,0 | 18,9 | 73 | 8 | 53 | 9 | 3 | 0,97 | 25 | 18 | 2,8 | 0,6 | 5 | 0,89 |
| SD | 2 | out_c_010 | 23 | 70 | 7,69E-06 | 7,28E-06 | 85 | 0,33 | 6,0 | 20,4 | 78 | 4 | 61 | 6 | 3 | 0,99 | 27 | 4 | 3,2 | 0,2 | 5 | 0,99 |
| SD | 3 | out_c_019 | 14 | 70 | 5,04E-06 | 3,79E-06 | 69 | 0,28 | 9,4 | 20,0 | 87 | 8 | 61 | 8 | 4 | 0,97 | 17 | 5 | 2,7 | 0,3 | 4 | 0,98 |
| LD | A | out_c_066 | 14 | 70 | 1,45E-05 | 1,83E-04 | 71 | 0,32 | 8,4 | 21,3 | 65 | 11 | 39 | 9 | 3 | 0,96 | 11 | 3 | 5,3 | 0,6 | 3 | 0,99 |
| SD | 4 | out_c_070 | 12 | 70 | 7,15E-06 | 8,26E-06 | 62 | 0,49 | 9,2 | 19,8 | 54 | 15 | 24 | 3 | 5 | 0,96 | 22 | 18 | 2,0 | 0,5 | 3 | 0,95 |
| SD | 5 | out_c_074 | 11 | 70 | 1,33E-05 | 1,33E-05 | 62 | 0,49 | 9,3 | 27,0 | 46 | 25 | 24 | 4 | 5 | 0,92 | nc | nc | nc | nc | nc | nc |
| SD | 6 | out_c_078 | 13 | 79 | 7,49E-06 | 1,39E-05 | 119 | 0,29 | 5,4 | 22,3 | 118 | 6 | 56 | 11 | 3 | 0,96 | 40 | 10 | 1,9 | 0,2 | 5 | 0,97 |
| SD | 7 | out_c_082 | 12 | 90 | 4,51E-06 | 2,79E-06 | 89 | 0,30 | 4,1 | 16,9 | 77 | 7 | 41 | 4 | 4 | 0,98 | 33 | 7 | 2,4 | 0,2 | 4 | 0,98 |
| SD | 8 | out_c_086 | 12 | 90 | 5,95E-06 | 6,75E-06 | 114 | 0,28 | 3,3 | 17,5 | nd | nd | nd | nd | nd | nd | nd | nd | nd | nd | nd | nd |
| SD | 9 | out_t_012 | 22 | 90 | 3,29E-06 | 2,36E-06 | nd | nd | nd | nd | 95 | 10 | 43 | 9 | 3 | 0,96 | 60 | 12 | 2,4 | 0,3 | 5 | 0,95 |
| SD | 10 | out_c_094 | 13 | 90 | 4,90E-06 | 4,30E-06 | 103 | 0,28 | 4,2 | 18,1 | 112 | 1 | 64 | 1 | 3 | 1,00 | nc | nc | nc | nc | nc | nc |
| SD | 11 | out_c_098 | 13 | 90 | 7,01E-06 | 8,37E-06 | 91 | 0,36 | 4,4 | 18,2 | 78 | 18 | 42 | 8 | 5 | 0,90 | 41 | 6 | 2,8 | 0,2 | 4 | 0,99 |
| SD | 12 | out_c_102 | 16 | 85 | 5,25E-06 | 4,80E-06 | 94 | 0,33 | 5,1 | 18,8 | 84 | 2 | 57 | 2 | 3 | 1,00 | na | na | na | na | na | na |
| LD | B | out_c_150 | 21 | 55 | 4,18E-06 | 6,66E-06 | 74 | 0,35 | 7,8 | 22,2 | 120 | 1 | 48 | 1 | 3 | 1,00 | nc | nc | nc | nc | nc | nc |
| SD | 13 | out_c_152 | 27 | x | nd | nd | 122 | 0,30 | 1,2 | 18,1 | 102 | 12 | 41 | 5 | 4 | 0,97 | na | na | na | na | na | na |
| LD | C | out_c_198 | 34 | 120 | 4,25E-06 | 1,49E-05 | 129 | 0,29 | 1,9 | 19,6 | 117 | 4 | 51 | 3 | 5 | 0,99 | na | na | na | na | na | na |
| SD | 14 | out_c_209 | 13 | 110 | 3,49E-06 | 4,11E-06 | 155 | 0,25 | 2,1 | 18,2 | 141 | 10 | 50 | 3 | 5 | 0,99 | na | na | na | na | na | na |
| LD | 15 | out_c_212 | 19 | 116 | 3,06E-06 | 2,45E-06 | 131 | 0,26 | 2,7 | 20,3 | 105 | 12 | 45 | 7 | 4 | 0,96 | na | na | na | na | na | na |





Table 2. General physical and biogeochemical characteristics for the 3 selected areas presented by columns: (1) western Melanesian Archipelago (WMA), eastern Melanesian Archipelago (EMA), western gyre (WGY) with the corresponding stations chosen; (2) Mean or standard deviation (SD); (3) MLD: Mixed layer depth (m); (4) EZD: Euphotic zone depth (m); 
5 (5) $K_z$: Mean 40-200 m vertical eddy diffusivity ($m^2$ $d^{-1}$); (6) DCMD: Deep chl a maximum depth; (7) DCMC: Deep chl a maximum concentration = chl a concentration at the DCMD (mg $m^{-3}$); (8 and 9) Ichl a, Integrated (0-70 m) and (0-200 m) total chl a concentration (mg $m^{-2}$); (10-13) $D_x$, $S_x$: characteristics of nutriclines (depths in m where $NO_3$ or $PO_4$ reaches zero and slopes in µmol $m^{-4}$). QL: Quantification limit; na: nutrient above QL at surface.

|  |  | MLD (dbar) | EZD (dbar) | $K_Z$ ($m^2$ $d^{-1}$) | DCMD (m) | Chl a DCMC (mg $m^{-3}$) | I 0-70m (mg $m^{-2}$) | I 0-200m (mg $m^{-2}$) | Nitracline $D_{NO3}$ (m) | $S_{NO3}$ (µmol $m^{-4}$) | Phosphacline $D_{PO4}$ (m) | $S_{PO4}$ (µmol $m^{-4}$) |
|---|---|---|---|---|---|---|---|---|---|---|---|---|
| WMA (SD 1,2,3 LD A) | Mean | 16 | 70 | 0,79 | 78 | 0,33 | 7,2 | 20,2 | 76 | 53 | 20 | 3,5 |
|  | SD | 5 | 0 | 0,34 | 10 | 0,05 | 2,1 | 1,0 | 10 | 10 | 7 | 1,2 |
| EMA (SD 6,7,9,10) | Mean | 15 | 87 | 0,44 | 104 | 0,29 | 4,6 | 19,1 | 100 | 51 | 44 | 2,2 |
|  | SD | 5 | 6 | 0,15 | 15 | 0,01 | 0,7 | 2,8 | 18 | 11 | 14 | 0,3 |
| EGY (SD 13,14,15 LD C) | Mean | 23 | 115 | 0,31 | 134 | 0,28 | 2,0 | 19,0 | 116 | 47 | na | na |
| 10 | SD | 9 | 5 | 0,05 | 14 | 0,03 | 0,6 | 1,1 | 18 | 5 | na | na |



Table 3. Mean integrated 0-70 m C, N, and P pools (mol m$^{-2}$) during the OUTPACE cruise (austral summer period) for the 3 selected areas: western Melanesian Archipelago (WMA), eastern Melanesian Archipelago (EMA), and western gyre (WGY). Dissolved inorganic (DI), dissolved organic (DO) and particulate organic (PO), C, N and P, respectively.

| | | | DIC | DOC | POC | DIN | DON | PON | DIP | DOP | POP |
|---|---|---|---|---|---|---|---|---|---|---|---|
| Austral summer | WMA | Mean | 141,2 | 5,07 | 0,21 | 0,000 | 0,392 | 0,035 | 0,0040 | 0,0112 | 0,0019 |
| | | SD | 0,3 | 0,12 | 0,02 | 0,000 | 0,036 | 0,004 | 0,0020 | 0,0018 | 0,0002 |
| | EMA | Mean | 141,6 | 5,22 | 0,22 | 0,000 | 0,370 | 0,031 | 0,0011 | 0,0117 | 0,0018 |
| | | SD | 0,6 | 0,07 | 0,02 | 0,001 | 0,017 | 0,002 | 0,0011 | 0,0010 | 0,0001 |
| | EGY | Mean | 141,9 | 5,35 | 0,09 | 0,000 | 0,378 | 0,015 | 0,0101 | 0,0136 | 0,0010 |
| | | SD | 0,4 | 0,08 | 0,01 | 0,000 | 0,045 | 0,001 | 0,0012 | 0,0020 | 0,0001 |



Table 4. Sediment trap data. Minimum, maximum and mean values of particulate matter and swimmer mass fluxes, C, N, P and Redfield ratio (RR) from particulate matter and swimmers (Zoo). * P calculated from the RR with adding the P measured in the supernatant. Last column: particulate matter and swimmers mass flux ratio. MA: Melanesian Archipelago, WGY: western SP gyre.

| | | Particulate matter | Swimmer | POC | PON | POP | RR 106/N/P proportion | | | Zoo-C | Zoo-N | Zoo-P | Zoo-P* | RR 106/N/P proportion | | | * | Zoo/Particulate |
| | | mgDW d⁻¹ | mgDW d⁻¹ | μmol m⁻² d⁻¹ | | | C | N | P | μmol m⁻² d⁻¹ | | | | C | N | P | P | mass flux ratio |
|---|---|---|---|---|---|---|---|---|---|---|---|---|---|---|---|---|---|---|
| MA | Min | 14 | 69 | 241 | 22 | 0,6 | 106 | 9,8 | 0,3 | 2 994 | 492 | 4 | 22 | 106 | 12,4 | 0,0 | 0,2 | 0,7 |
| (N=8) | Max | 122 | 403 | 3 084 | 395 | 68,9 | 106 | 15,4 | 2,9 | 11 742 | 1 653 | 26 | 55 | 106 | 20,2 | 0,6 | 1,1 | 26,0 |
| | Mean | 48 | 219 | 1 092 | 136 | 18,5 | 106 | 12,7 | 1,2 | 6 903 | 961 | 14 | 36 | 106 | 15,8 | 0,2 | 0,7 | 9,7 |
| WGY | Min | 7 | 24 | 138 | 20 | 0,4 | 106 | 11,2 | 0,2 | 609 | 75 | 2 | 5 | 106 | 13,1 | 0,2 | 0,5 | 1,4 |
| (N=8) | Max | 28 | 148 | 385 | 59 | 2,1 | 106 | 20,4 | 0,8 | 4 552 | 1 067 | 16 | 28 | 106 | 25,2 | 0,5 | 0,9 | 7,1 |
| | Mean | 17 | 76 | 266 | 41 | 1,1 | 106 | 16,6 | 0,5 | 2 330 | 473 | 8 | 15 | 106 | 19,9 | 0,3 | 0,7 | 4,4 |



Table 5. Molar fraction of $CO_2$ ($X_{CO2}$) in dry air measured at SMO station Tutuila (American Samoa, Lat 14.247° S, Lon 170.564° W, see Fig. 1; Source: NOAA/ESRL) and derived atmospheric $p_{CO2}$ ($p_{CO2}^{atm}$). Mean values for the carbonate system, measured in the mixed layer depth (MLD) during the OUTPACE cruise (summer conditions), measured at 70 m depth (estimated winter conditions) and NDP-094 climatological data (Takahashi et al., 2014). Oceanic climatological $p_{CO2}$ ($p_{CO2}^{oc}$) are given for different estimations of winter temperature (SST from NDP-094, mean T at 70 m depth from OUTPACE, SST from MODIS Aqua). CT: conservative temperature, $S_p$: practical salinity, nDIC: normalized dissolved inorganic carbon, nAlk: normalized alkalinity, SST: sea surface temperature; SSS: sea surface salinity). WMA: western Melanesian Archipelago, EMA: eastern Melanesian Archipelago, and WGY: western SP gyre sampled during the OUTPACE cruise.

| SMO station TUTUILA | Monthly mean $X_{CO2}$ (March 2015) | ppm | 398,4 | | | | | |
| | Annual mean $X_{CO2}$ (Jul 2014-Jul 2015) | ppm | 397,3 (SD = 0,8) | | | | | |
| | | | WMA | | EMA | | WGY | |
| | Unit | | Mean | SD | Mean | SD | Mean | SD |
| | Monthly mean $p_{CO2}^{atm}$ (March 2015) | µatm | 383,0 | | 382,7 | | 382,5 | |
| Mean values measured in the MLD (austral summer conditions, March 2015) | CT | ℃ | 28,9 | 0,3 | 29,3 | 0,3 | 29,5 | 0,4 |
| | $S_p$ | | 35,0 | 0,1 | 35,1 | 0,2 | 35,1 | 0,1 |
| | nDIC | µmol kg⁻¹ | 1974,6 | 9,5 | 1983,9 | 8,6 | 1993,6 | 3,2 |
| | nAlk | µmol kg⁻¹ | 2333,7 | 1,9 | 2343,1 | 8,0 | 2347,7 | 6,1 |
| | $nP_{CO2}^{oc}$ | µatm | 366 | 11 | 376 | 8 | 390 | 6 |
| Mean values at 70 m (estimated austral winter MLD conditions) | CT | ℃ | 25,3 | 0,2 | 24,8 | 0,8 | 26,2 | 0,9 |
| | $S_p$ | | 35,5 | 0,0 | 35,7 | 0,0 | 35,7 | 0,0 |
| | nDIC | µmol kg⁻¹ | 2007,5 | 3,0 | 2009,6 | 9,6 | 2008,9 | 3,7 |
| | nAlk | µmol kg⁻¹ | 2332,4 | 5,0 | 2344,1 | 6,5 | 2350,8 | 2,7 |
| | $nP_{CO2}^{oc}$ | µatm | 372 | 10 | 355 | 15 | 364 | 8 |
| Austral winter (July 2014) temperature at surface from MODIS Aqua | SST | ℃ | 24,9 | 0,2 | 24,2 | 0,7 | 26,5 | 0,2 |
| Mean climatological austral winter values at surface (from Takahashi et al., 2014) | SST | ℃ | 23,9 | 0,0 | 24,4 | 0,1 | 25,1 | 0,1 |
| | SSS | | 35,5 | 0,0 | 35,5 | 0,0 | 35,6 | 0,1 |
| | DIC | µmol kg⁻¹ | 2006,4 | 0,7 | 2000,9 | 3,0 | 2004,7 | 9,9 |
| | Alk | µmol kg⁻¹ | 2335,4 | 0,2 | 2333,6 | 1,7 | 2343,4 | 8,6 |
| | $P_{CO2}^{oc}$ at surface | µatm | 344 | 1 | 345 | 4 | 349 | 7 |
| Mean climatological austral winter $P_{CO2}^{oc}$ at surface calculated with different temperatures | T from OUTPACE, mean CT at 70 m depth | µatm | 366 | | 353 | | 368 | |
| | SST from MODIS Aqua | µatm | 361 | | 344 | | 371 | |



Table 6. 0-70 m upper layer comparative C, N, P daily budgets in the 3 selected areas (western Melanesian Archipelago (WMA), eastern Melanesian Archipelago (EMA), and western gyre (WGY)) sampled during the OUTPACE cruise ($\mu$mol m$^{-2}$ d$^{-1}$).

| | | WMA | | | EMA | | | EGY | | |
|---|---|---|---|---|---|---|---|---|---|---|
| | | C | N | P | C | N | P | C | N | P |
| INPUT | Dissolved inorganic turbulent diffusion | 426 | 46 | 2,8 | 198 | 0 | 1,0 | 200 | 0 | 0,0 |
| | Atmospheric $CO_2$ exchange or $N_2$ fixation | 1675 | 642 | negligible | 825 | 452 | negligible | 0 | 41 | negligible |
| OUTPOUT | Particulate organic settling | -1092 | -136 | -18,5 | -1092 | -136 | -18,5 | -266 | -41 | -1,1 |
| | Particulate organic turbulent diffusion | -29 | -4 | -0,2 | -16 | -1 | -0,1 | 2 | 0 | 0,0 |
| | Dissolved organic turbulent diffusion | -121 | -8 | -0,7 | -58 | -6 | -0,3 | -21 | 0 | 0,0 |
| BUDGETS | | 859 | 540 | -16,6 | -143 | 309 | -17,9 | -85 | 0 | -1,1 |





Table 7. Estimated temporal evolution of surface biogeochemical properties between austral winter (up) and summer (below) in µmol L$^{-1}$ presented by columns: (1) Mean measurements at 70 m depth during OUTPACE were considered to represent the homogeneous upper water column (0-70 m) variables and initial winter conditions (i.e. conditions in July 2014). The summer conditions were those observed during the OUTPACE cruise (March 2015). Δ represents the summer – winter conditions differences. (2) Selected area: western Melanesian Archipelago (WMA), eastern Melanesian Archipelago (EMA) or western gyre (WGY). (3) Mean or standard deviation (SD), (4 to 15) Dissolved inorganic (DI), dissolved organic (DO), particulate organic (PO), and total (T), C and N and P, respectively. All variables were normalized to the mean absolute salinity measured at 70 m depth to discard evolution due to evaporation/precipitation.

| | | | nDIC | nDOC | nPOC | nTC | nDIN | nDON | nPON | nTN | nDIP | nDOP | nPOP | nTP |
|---|---|---|---|---|---|---|---|---|---|---|---|---|---|---|
| Austral winter | WMA | Mean | 2055,1 | 67,5 | 1,6 | 2124,2 | 0,00 | 5,06 | 0,38 | 5,44 | 0,142 | 0,125 | 0,022 | 0,289 |
| | | SD | 3,0 | 4,4 | 0,5 | | 0,00 | 0,38 | 0,06 | | 0,030 | 0,024 | 0,005 | |
| | EMA | Mean | 2057,5 | 68,9 | 2,3 | 2128,7 | 0,03 | 4,93 | 0,34 | 5,30 | 0,059 | 0,140 | 0,021 | 0,220 |
| | | SD | 10,2 | 2,0 | 0,4 | | 0,04 | 0,27 | 0,05 | | 0,046 | 0,028 | 0,004 | |
| | EGY | Mean | 2056,2 | 74,2 | 1,4 | 2131,8 | 0,01 | 5,52 | 0,24 | 5,77 | 0,142 | 0,182 | 0,015 | 0,339 |
| | | SD | 4,3 | 2,3 | 0,2 | | 0,01 | 0,74 | 0,01 | | 0,020 | 0,037 | 0,001 | |
| Austral summer | WMA | Mean | 2018,1 | 77,6 | 4,3 | 2100,0 | 0,00 | 6,05 | 0,68 | 6,73 | 0,000 | 0,163 | 0,032 | 0,195 |
| | | SD | 10,0 | 2,2 | 0,3 | | 0,00 | 0,58 | 0,12 | | 0,000 | 0,029 | 0,004 | |
| | EMA | Mean | 2027,5 | 78,2 | 4,5 | 2110,1 | 0,00 | 5,72 | 0,59 | 6,31 | 0,000 | 0,180 | 0,031 | 0,211 |
| | | SD | 8,8 | 1,1 | 1,1 | | 0,00 | 0,39 | 0,15 | | 0,000 | 0,018 | 0,005 | |
| | EGY | Mean | 2037,6 | 79,2 | 1,2 | 2117,9 | 0,00 | 5,50 | 0,21 | 5,71 | 0,150 | 0,190 | 0,013 | 0,353 |
| | | SD | 3,4 | 1,1 | 0,2 | | 0,00 | 0,70 | 0,04 | | 0,014 | 0,017 | 0,002 | |
| Δ | WMA | | -37,0 | 10,1 | 2,6 | -24,2 | 0,00 | 1,00 | 0,29 | 1,29 | -0,142 | 0,037 | 0,010 | -0,095 |
| | EMA | | -30,0 | 9,3 | 2,2 | -18,6 | -0,03 | 0,79 | 0,25 | 1,01 | -0,059 | 0,040 | 0,010 | -0,009 |
| | EGY | | -18,7 | 5,0 | -0,2 | -13,9 | -0,01 | -0,02 | -0,03 | -0,06 | 0,008 | 0,008 | -0,002 | 0,014 |





Table 8. Estimated temporal evolution of (0-70 m) biogeochemical inventories between austral winter and summer in mmol m$^{-2}$ presented by columns: (1) Mean measurements at 70 m depth during OUTPACE were considered to represent the homogeneous upper water column (0-70 m) variables and initial winter conditions (i.e. conditions in July 2014). The summer conditions were those observed during the OUTPACE cruise (March 2015). Δ represents the summer – winter

5 conditions differences. (2) Selected area: western Melanesian Archipelago (WMA), eastern Melanesian Archipelago (EMA) or western gyre (WGY). (3) Mean or standard deviation (SD), (4 to 15) Dissolved inorganic (DI), dissolved organic (DO), particulate organic (PO), and total (T), C and N and P, respectively. All variables were normalized to the mean absolute salinity measured at 70 m depth to discard evolution due to evaporation/precipitation.

| | | | nDIC | nDOC | nPOC | nTC | nDIN | nDON | nPON | nTN | nDIP | nDOP | nPOP | nTP |
|---|---|---|---|---|---|---|---|---|---|---|---|---|---|---|
| Δ (summer - winter) | WMA | Mean | -1563 | 391 | 102 | -1070 | 0,0 | 40,8 | 8,5 | 49 | -5,9 | 2,4 | 0,4 | -3,1 |
| | | SD | 209 | 307 | 33 | | 0,0 | 35,6 | 4,1 | | 2,1 | 1,8 | 0,4 | |
| | EMA | Mean | -1355 | 445 | 55 | -855 | -1,7 | 28,0 | 7,5 | 34 | -3,0 | 1,8 | 0,3 | -0,9 |
| | | SD | 713 | 139 | 26 | | 3,1 | 19,0 | 3,7 | | 3,2 | 2,0 | 0,3 | |
| | EGY | Mean | -659 | 220 | -8 | -448 | -0,3 | -4,6 | -1,5 | -6 | 0,1 | 1,0 | -0,1 | 1,1 |
| | | SD | 298 | 162 | 14 | | 0,9 | 51,5 | 1,3 | | 1,4 | 2,6 | 0,1 | |





**Figures caption**

Fig. 1. Transect of the OUTPACE cruise (18 Feb. 3 Apr. 2015) superimposed on a bathymetry map (GEBCO_2014 grid) of the Western Tropical South Pacific Ocean. The two types of stations, short duration and long duration, are indicated together with the stations chosen to represent 3 selected areas: the western Melanesian Archipelago (WMA in dark green), the eastern Melanesian Archipelago (EMA in light green) and the western SP gyre (WGY in blue). SMO station Tutuila (American Samoa, Lat 14.247° S, Lon 170.564° W).

Fig. 2. Monthly mean mixed layer depth (MLD) against month in climatology (a, d, g), sea surface temperature (SST) (b, e, h) and chl $\underline{a}$ (c, f, i) against months from July 2014 to July 2015, respectively, for (a, b, c) the western Melanesian Archipelago (WMA), (d, e, f) the eastern Melanesian Archipelago (EMA) and (g, h, i) the western SP gyre (WGY).

Fig. 3. Vertical profiles of (a) conservative temperature (CT in °C), (b) absolute salinity (SA in g $kg^{-1}$), (c) potential density (Sigma in kg $m^{-2}$), (d) dissolved oxygen ($O_2$ in µmol $kg^{-1}$), (e) apparent oxygen utilization (AOU in µmol $kg^{-1}$) and (f) chl $\underline{a}$ (mg $m^{-3}$) versus depth (0-200 m) in the 3 distinct areas sampled during the OUTPACE cruise: the western Melanesian Archipelago (WMA in dark green), the eastern Melanesian Archipelago (EMA in light green) and the western SP gyre (WGY in blue).

Fig. 4. Vertical profiles of (a) dissolved inorganic carbon (DIC), (b) total alkalinity (TA), (c) normalized dissolved inorganic carbon (nDIC) and (d) normalized total alkalinity (nTA) against depth (m) for the 3 distinct areas sampled during the OUTPACE cruise: the western Melanesian Archipelago (WMA in dark green), the eastern Melanesian Archipelago (EMA in light green) and the western SP gyre (WGY in blue).

Fig. 5. Vertical profiles of normalized (n) C, N, P data against depth (m). Dissolved inorganic (DI), dissolved organic (DO) and particulate organic (PO) C (a, d, g), N (b, e, h) and P (c, f, i), respectively, in µmol $L^{-1}$, for the 3 distinct areas sampled during the OUTPACE cruise: the western Melanesian Archipelago (WMA in dark green), the eastern Melanesian Archipelago (EMA in light green) and the western SP gyre (WGY in blue).

Fig. 6. Vertical profiles of (a) primary production (PP rate in nmolC $L^{-1}$ $d^{-1}$), (b) $N_2$ fixation rate (nmolN $L^{-1}$ $d^{-1}$) and dissolved inorganic phosphate turnover times ($T_{DIP}$ in days on log scale) against depth for the 3 distinct areas sampled during the OUTPACE cruise: the western Melanesian Archipelago (WMA in dark green), the eastern Melanesian Archipelago (EMA in light green) and the western SP gyre (WGY in blue).

Fig. 7. C, N, and P estimated budgets in the 0-70 m water column during the 8-month period between deep convection in July 2014 (austral winter) and strong stratification in March 2015 (austral summer) for the 3 distinct areas sampled during the OUTPACE cruise: the western Melanesian Archipelago (WMA, top), the eastern Melanesian Archipelago (EMA, middle) and the western SP gyre (WGY, bottom). C budgets (7a, 7d, 7g), N budgets (7b, 7e, 7h) and P budgets (7c, 7f, 7i). Dissolved inorganic (DI), dissolved organic (DO), and particulate organic (PO) C, N, and P fluxes are considered, respectively. Atmospheric exchanges limited to $CO_2$ penetration and $N_2$ fixation are indicated. All fluxes are expressed in mmol $m^{-2}$ (of elemental C, N, and P, respectively) with arrows indicating direction (input or output). The 2 numbers for the particulate fluxes correspond to fluxes by turbulent diffusion (above) and particle settling (below). Estimated accumulation rates for the same period are indicated inside the boxes.



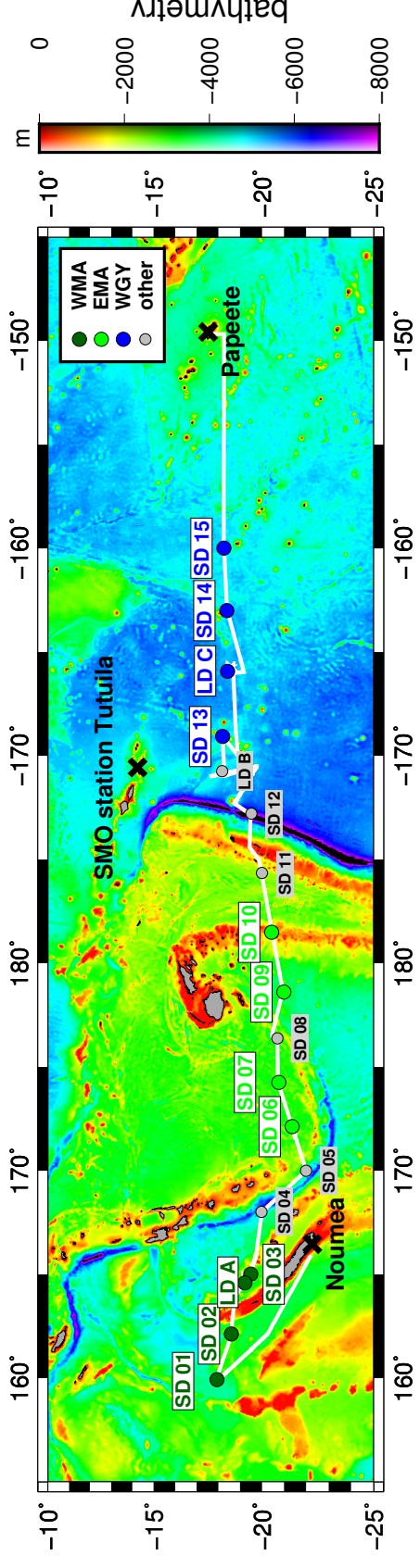

Figure 1



Figure 2





Figure 3





Figure 4





Figure 5



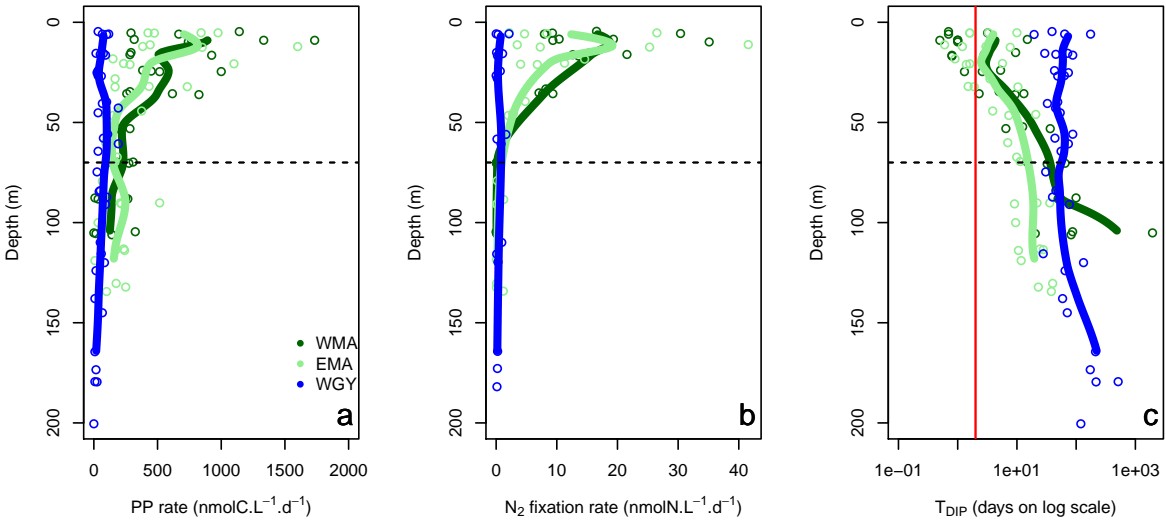

Figure 6





Figure 7