# Peer review of "Nutrient availability and the ultimate control of the biological carbon pump in the Western Tropical South Pacific Ocean"

_Biogeosciences, 2017_

## Referee Comment (RC1) · Anonymous Referee #1 · 31 Jan 2018

On the one hand, this ms. deals with an impressive dataset, but in the end, the main conclusions (importance of iron, importance of zooplankton vertical migration for export) are conjectured from indirect evidence, not from processes/data studied by the authors on the cruise.

There are several grave deficiencies in the data treatment and interpretations.

1. Why should it be justified to obtain "winter values" of properties by taking the 70m values as representative? These are minimum estimates at best given that in almost all profiles the chl-a maximum is below 70m.

2. How can matter, recovered from sediment traps, be taken at face value and trap

biases be not considered? Or, even better, corrected for! This issue has been exten-sively discussed: Gardener 2000 in The changing ocean carbon cycle, Hanson, Duck-low, Field, eds., Cambridge Univ. Press, p 240-281; Buesseler et al. 2007, J.Mar.Res. 65, 345-416). Solubilization into collection cup supernatants is (almost) not consid-ered at all but this is important especially in shallow traps, see Kähler and Bauerfeind (2001), Limnol. Oceanogr. 46, 719-723; Antia (2005) Biogeosci. Discussions 275-302. In the case of phosphorus, supernatants have been measured but assigned 100% to swimmers! What about P in the passive flux? DOC could obviously not be measured (formalin) and DON was not, or is not reported. Hence almost all P and much N and C "export" by particles is missed in the trap data. Also one of the traps is used in the balances of two regions (Fig. 7).

3. Zooplankton vertical migration is conjectured to be the one important export mech-anism with only implied evidence (ADCP data not shown). But to present swimmers attracted to the sediment trap as evidence of vertical migration ("Zoo/Particulate mass flux ratio" in Tab.4) or "mean carbon export by swimmers" (p.13 line 23) is downright nonsense. Swimmers killed in the trap cannot move upward again. And of what should swimmers be representative? Certainly not of a vertical flux.

4. New production and primary production are not neatly distinguished in the text (e.g. p.13 line37; p13 line 19). Most PP is recycled in these environments thus PP cannot explain the drawdown of DIC.

5. The ms. was to me difficult to read because of abounding abreviations and numbers in the text, repeated checks were necessary whether the authors refer to their own data or speculate. No story is told convincingly. But my rejection of the ms. is mainly based on points 2 and 3 above.

---

## Referee Comment (RC2) · Anonymous Referee #2 · 31 Jan 2018

This manuscript reports upper water column biogeochemical observations from a cruise in the western tropical South Pacific. The cruise programme had a particularly novel finding: elevated rates of nitrogen fixation throughout the Melanesian Archipelago. These results have already been reported: briefly in Bonnet et al. (2017), and in detail in a companion manuscript as part of the cruise special issue in Biogeosciences Discussions (Bonnet et al., in review). The novel contribution of the present manuscript is that an upper water column carbon budget for the region, built from observations on the cruise and longer time series datasets, appears to require an additional fixed nitrogen source other than seasonal entrainment of waters below the mixed layer. They interpret this deficit as originating from the enhanced nitrogen fixation observed

(Bonnet et al., 2017; Bonnet et al., in review). Provided all carbon system calculations are correct—which is not my area of expertise—I think the manuscript is of significant interest to the scientific community. I do however have a number of comments that should be addressed.

Main comments:

Clarity of writing. The manuscript requires a large number of corrections for English, sentence structure, and correct terminology. This will much improve the readability of the paper and make it more readily understandable. There were a number of instances where I had to read a paragraph more than once to work out what message the author was trying to convey (and not always with 100% success!). Therefore I would recommend going through the paper in detail with a native English scientist in order to check sentence structure and terminology for simplicity and clarity. I have made some suggestions in the specific comments section, but these are only examples; there are more throughout the manuscript.

The role of Fe. The manuscript argues for a primary role of Fe in enabling nitrogen fixers to become established in the Melanesian Archipelago, but then invoke phosphorus availability as the primary factor controlling the activity of nitrogen fixers. Accordingly, Fe is frequently stated as 'high' (or 'Fe-replete') in Melanesian Archipelago area. Where are the Fe data? The authors do give some values from a paper in review that appear high, however, since this is central to the manuscripts conclusions, some more details are needed to describe how Fe concentrations varied though surface waters of the three regions discussed.

Surface Fe supply from shallow hydrothermal vents is also mentioned, which is very interesting, however as I understand it none of these data are yet published. Regardless, whilst this could significantly influence the overall Fe budget and capacity of the Melanesian Archipelago system for nitrogen fixation, this does not preclude periodically or seasonally low levels of Fe in surface waters of the Melanesian Archipelago.

[Figure]

In other words, what evidence is there that Fe is at high, steady state value in this bio-geochemical province? Supplied Fe has a short lifetime in seawater because of rapid scavenging and concentrations following a point source supply diminish rapidly and require continuous inputs to lead to sustained high surface concentrations (such as under the Saharan dust plume in the tropical North Atlantic). Other Fe values reported by Moisander et al. (2011) in the vicinity of the region appear lower than the two values stated.

Sources of fixed nitrogen. Currently the authors use profiles of nitrate concentrations measured during their cruise and climatology of MLD to estimate nitrate entrainment during deeper wintertime mixing. From this they conclude that nitrate input to the surface mixed layer by this process is minimal. Indeed looking at the depth profiles of nitrate (nitracline ~70m depth) and MLD climatology (maximum mixed layer of ~75m) this would appear to be sound. However, in using climatological average mixed layer depths, the authors do not consider periodic entrainment of much deeper waters with much higher nitrate concentrations by transient storms (see, for example, general cyclone passages at: https://en.wikipedia.org/wiki/Tropical_cyclogenesis#/media/File:Global_tropical_cyclone_tracks-edit2.jpg). What is the role of periodic nitrate entrainment by these deep mixing events that are not characterised by average MLD climatology? It is worth noting that such storms would also entrain DIC into the mixed layer, as well as nitrate, and the net effect on carbon budgets might be low.

What is the mesoscale eddy activity in this region? These have been shown to supply significant N to other oligotrophic waters. Could eddies be supplying additional nitrate? See, for example, Falkowski et al. 1991; Oschlies and Garcon, 1998.

Additional sediment trap sample details. Some details with regards to the type of material found in the sediment traps could be valuable to support the 'mechanistic' discussion with respect to nitrogen fixation-fuelled carbon export. Were significant Trichodesmium colonies found in the sediment traps? Or diazotrophic diatoms? Or zooplankton? Partly related to this, are there any details about phytoplankton community structure determined from HPLC pigment analyses, flow cyctometry, or microscopy? I appreciate that all these details could be in another article in the special issue, but I cannot see it mentioned. More generally the manuscript should be very clear/explicit as to what the new data in this manuscript are, and what is has been published/is in review elsewhere.

More specific comments

ADCP data. These data are mentioned with regards to the zooplankton migration, with implications for sub-mixed layer carbon export. Can we see plots of the data? How were the reported zooplankton respiratory carbon losses to the sub-mixed layer calculated?

DOP and alkaline phosphatase activity. Figure 5 shows significant phosphate available as DOP, even in DIP-depleted Melanesian Archipelago waters. Could the P demand of nitrogen fixers be sustained by DOP access? Was alkaline phosphatase activity measured? Recent findings show that Fe/Zn is required for some dominant forms of alkaline phosphatase and so Fe/Zn availability could also control P access, in addition to N access via nitrogen fixation (Mahaffey et al., 2014; Browning et al., 2017i; Landolfi et al., 2015).

Relative influence of iron and nitrogen in the WGY region. Biologically accessible nitrogen is inferred to be the limiting resource to the overall phytoplankton community in the WGY (western gyre region) (with Fe being the limiting nutrient for diazotrophs). What is the evidence that nitrogen and iron are not directly co-liming to the overall phytoplankton community in this region (Saito et al., 2014; Browning et al., 2017ii—both studies through the edge of gyre boundaries)? In the study of Moisander et al. (2011), which was in proximity to the OUTPACE cruise region, a bottle enrichment experiment displayed Fe was serially limiting (following nitrogen) to the overall phytoplankton community, supporting this possibility.

Co-limitation. Co-limitation is not mentioned once in the manuscript but it seems central to the discussion. For instance, iron and phosphorus could co-limit nitrogen fixation (Mills et al., 2004), but iron could also co-limit phosphorus acquisition by the microbial community from the DOP pool, and iron and nitrogen could co-limit the overall phytoplankton community. These interactions and potential for feedbacks are pertinent when hypothesising a role of 'ultimate' limiting nutrients and potential for future change (both topics of this manuscript). To quote Moore et al. (2013; referenced in the manuscript): "Establishing the identity of a single ultimate limiting nutrient may thus be less relevant than understanding the controls on, and feedbacks pertaining to, any given process". Personally I would avoid conclusions/discussion that heavily refer to 'ultimate' limiting nutrients (e.g. Moore et al., 2013; Tyrrell, 1999) as this is more relevant to larger spatial-temporal scales than this dataset can be used for.

Figures. Figures are generally clear but some details to captions need to be added. For example, check that all vertical/horizontal lines in Figs 3–6 are defined in the figure caption (even if defined in the text). Figure 1 (map) is good for detail but not very good for placing the cruise in its wider geographic position. Could an inset map or similar with continents for geographic context be included in addition to the current map? See Bonnet et al. (in review in this special issue Fig 1).

Tables. Currently commas rather than decimal points are used in tables. This could be confusing for some readers. Please change.

Spelling/sentence structure/grammar: see first main point, below are some examples:
Pg 2 Line 3: 'deep Sea' → 'deep sea'

Pg 5 Line 41: allowing to draw first-order winter to summer seasonal variations → 'allowing us to draw first-order winter to summer seasonal variations'

Pg 6 Line 10: 'and 2, 3, and 3' → 'and 2, 3, and 4'?

Pg 8 line 23: 'Rapidly, seawater was collected in triplicates from the Niskin bottles in 2.3

L polycarbonates bottles at 6 depths (75 %, 54 %, 19 %, 10 %, 1 %, and 0.1 % surface irradiance levels), like for PP measurements' → 'As for PP measurements, seawater was rapidly collected in triplicate from the Niskin bottles in 2.3 L polycarbonates bottles at 6 depths (75 %, 54 %, 19 %, 10 %, 1 %, and 0.1 % surface irradiance levels).'

Pg 11, line 12: 'Otherwise, very low and improbable P contents were found in the swimmers' → Please be more specific with regards to 'very low' and 'improbable'

Pg. 11 line 37: 'large precipitation' → do you mean rainfall?

Pg 12 line 41: 'Else we need. . .' → 'Otherwise we need..'

Pg 18 Line 17: 'from an iron limitation in the east' → 'from probable iron limitation in the east'; currently no iron data is given to rule out other potential controls.

Pg. 18 Line 28: 'Furthermore, both diazotrophy and denitrification are known to undergo drastic alterations due to climate change.' → References needed to back up this statement?

References

Browning, T.J., Achterberg, E.P., Yong, J.C., Rapp, I., Utermann, C., Engel, A. and Moore, C.M., 2017i. Iron limitation of microbial phosphorus acquisition in the tropical North Atlantic. Nature Communications, 8.

Browning, T.J., Achterberg, E.P., Rapp, I., Engel, A., Bertrand, E.M., Tagliabue, A. and Moore, C.M., 2017ii. Nutrient co-limitation at the boundary of an oceanic gyre. Nature, 551(7679), p.242.

Falkowski, P.G., Ziemann, D., Kolber, Z. and Bienfang, P.K., 1991. Role of eddy pumping in enhancing primary production in the ocean. Nature, 352(6330), p.55.

Landolfi, A., Koeve, W., Dietze, H., Kähler, P. and Oschlies, A., 2015. A new perspective on environmental controls of marine nitrogen fixation. Geophysical Research Letters, 42(11), pp.4482-4489.

Mahaffey, C., Reynolds, S., Davis, C.E. and Lohan, M.C., 2014. Alkaline phosphatase activity in the subtropical ocean: insights from nutrient, dust and trace metal addition experiments. Frontiers in Marine Science, 1, p.73.

Mills, M.M., Ridame, C., Davey, M., La Roche, J. and Geider, R.J., 2004. Iron and phosphorus co-limit nitrogen fixation in the eastern tropical North Atlantic. Nature, 429(6989), pp.292-294.

Moisander, P.H., Zhang, R., Boyle, E.A., Hewson, I., Montoya, J.P. and Zehr, J.P., 2012. Analogous nutrient limitations in unicellular diazotrophs and Prochlorococcus in the South Pacific Ocean. The ISME journal, 6(4), pp.733-744.

Moore, C.M., Mills, M.M., Achterberg, E.P., Geider, R.J., LaRoche, J., Lucas, M.I., McDonagh, E.L., Pan, X., Poulton, A.J., Rijkenberg, M.J. and Suggett, D.J., 2009. Large-scale distribution of Atlantic nitrogen fixation controlled by iron availability. Nature Geoscience, 2(12), pp.867-871.

Moore, J.K. and Doney, S.C., 2007. Iron availability limits the ocean nitrogen inventory stabilizing feedbacks between marine denitrification and nitrogen fixation. Global Biogeochemical Cycles, 21(2).

Saito, M.A., McIlvin, M.R., Moran, D.M., Goepfert, T.J., DiTullio, G.R., Post, A.F. and Lamborg, C.H., 2014. Multiple nutrient stresses at intersecting Pacific Ocean biomes detected by protein biomarkers. Science, 345(6201), pp.1173-1177.

Oschlies, A. and Garcon, V., 1998. Eddy-induced enhancement of primary production in a model of the North Atlantic Ocean. Nature, 394(6690), pp.266-269.

---

## Referee Comment (RC3) · Anonymous Referee #3 · 8 Feb 2018

Moutin et al.'s study investigates the regulation of the ocean C, N and P fluxes in the western tropical South Pacific Ocean, a key oligotrophic oceanic region for nitrogen fixation. They combine a large new dataset collected during the OUTPACE campaign (February-April 2015) with climatologies of upper water properties. The OUTPACE cruise followed a 4000 km longitudinal transect going from the Melanesian Archipelago (MA) to Papeete (French Polynesia). Comparing seasonal trends of C, N, and P mass balances for 3 main areas of the WTSP, they find that (1) the MA is a net sink of atmospheric $CO_2$, mainly caused by the soft-tissue biological pump; (2) the MA biological pump results from both high rate of N2 fixation fuelling export production and meso-zooplankton diel vertical migration; and (3) MA N2 fixation is essentially controlled by

phosphate availability.

I think the tackled questions and presented results of this study are really interesting and important and deserve publication. There are lots of uncertainties on the role of nitrogen fixers in fuelling ocean production and export in oligotrophic area especially of the Pacific Ocean due to lack of observations. This region is particular important as covering a large area and presenting high rates of N2 fixation. I thus strongly recommend this work for publication. I have however the following main comments that need to be taken into consideration.

Main comments

Paper presentation - At the moment the results and the argument for the role of nitrogen fixation and mesozooplankton vertical migration for the atmospheric CO2 sink of the Melanesian Archipelago region are a bit convoluted. While it is nice to follow the steps of thought of the authors on how they come about to find these links, this is done a step too far: for instance the second half of section 4.3 more or less repeats what is said in section 4.2. I recommend the authors to reorganise the results and discussion to be a lot more concise. This will help following the argument and strengthening the case of the paper. To help make a stronger case for the paper, I wonder also if it would be possible to add a diagram that visualises the main processes occurring in the region (N2 fixation, vertical migration of zooplankton, CO2 uptake . . .).

Effect of zooplankton vertical migration - The current manuscript presents the vertical migration of mesozooplankton as an explanation for the missing sink in the upper C budget at WMA. It would be nice to quantify this process either based on other technique or by simply doing a mass balance (which the authors are shying away).

MA net sink of atmospheric CO2 - It would be good to quantify the net sink in pCO2 and compare it with other estimates and with other regions.

Minor comments

P2, L3: Confusing sentence. In my mind mineral refers to carbonate and silicate which is at odd in association with the soft-tissue pump which refers to organic carbon.

P3, L35: Can you give slightly more information of the climatology of de Boyer-Montegut et al. (2004)? Mainly the type of collected data and coverage.

P6, L39: Say in the Figure 2 caption, what the red lines refer to.

P9, L27: Need to specify that shallow nitracline indicates oligotrophic waters.

P12, L35: Can you add the sum of the MA regions so that we can see that the MA area is a net sink of atmospheric $CO_2$? P12, L40: Little bit misleading as the Table 6 and main text do not use the same unit. Can you add in the text the numbers in mol m-2 d-1 as well?

P18, L40: Can you stipulate here if the source of P changes with climate change, how this might affect N2 fixation, zooplankton migration and $CO_2$ sink?

---

## Author Comment (AC1) · 19 Mar 2018

**On the one hand, this ms. deals with an impressive dataset, but in the end, the main conclusions (importance of iron, importance of zooplankton vertical migration for export) are conjectured from indirect evidence, not from processes/data studied by the authors on the cruise.**
**There are several grave deficiencies in the data treatment and interpretations.**

We thank Referee #1 for remarking on the 'impressive dataset' the ms deals with, and for taking the time to read it and providing his/her thoughts and comments. In the following paragraphs, the original review comments from Referee #1 are in bold and our own responses are interspersed in normal characters. Our main conclusion in the paper is that the discrepancies between the depth of the nitracline and phosphacline, together with evidence of deep mixing at 70 m, allow excess P to annually reach the upper surface, support nitrogen fixation, and induce a significant biological soft tissue pump in the MA. Based on the available iron data in the area, we can consider that the WMA is replete in iron, while the gyre is iron depleted. The importance of phosphate in the WTSP and iron in the gyre is not a main conclusion of this paper. It was the conclusion of a previously published study (Moutin et al., 2008). This conclusion is extended in the present paper. Because this was the starting point of the narrative of this study, it was necessary to recall the critical role of iron and phosphate in the Abstract, and to discuss specifically recent advances on this subject (section 4.4). Again, our main conclusion concerns the local origin of the excess-P at an annual time scale and subsequent processes, such as the role of $N_2$ fixation in nutrient availability and the biological carbon pump. The importance of zooplankton vertical migration is suggested by our data, but not for export (as stipulated by Referee #1). It is suggested to explain part of the missing DIC in the upper surface carbon budgets. Nevertheless, we recognize that some sentences may prove confusing and apologize for the lack of precision in the section 'Settling particulate matter mass, C, N and P flux measurements'. A detailed protocol is added at the end of this response. As requested by Referees #2 and # 3, we will provide figures of the ADCP data showing vertical migrations. Because it seems to be a point of contention and because it is for us a secondary subject that will distract from our main conclusion, this part will be has been considerably shortened in the new version of the ms. Please note that we only needed to delete one sentence in the Abstract [the following: "We also suggest that mesozooplankton diel vertical migration plays a dominant role in the transfer of carbon from the upper surface to deeper water in the MA. ": 27/422 words] to answer Referee #1's main arguments in favour of rejection (point 5). The Discussion will be considerably shortened.

Moutin, T., Karl, D. M., Duhamel, S., Rimmelin, P., Raimbault, P., Van Mooy, B. A. S., and H. Claustre. 2008. Phosphate availability and the ultimate control of new nitrogen input by nitrogen fixation in the tropical Pacific Ocean, Biogeosciences, 5, 95-109.

**1. Why should it be justified to obtain "winter values" of properties by taking the 70m values as representative?**
"Winter values" of properties are essential to draw to establish simple budgets. We understand the question as: "..will it be possible to consider 70 m values (70 m being the maximum mixed layer depths (MLD) calculated for the whole area) as representative of homogenized 0-70 m winter conditions?"

We only have satellite and some climatological data to estimate winter conditions in this largely under-sampled ocean. Therefore, we firstly validated satellite measurements during our survey in March (SST as an example below)

|  | WMA | EMA | EGY |
|---|---|---|---|
| SST(°C) March 2015 | 28,8 ± 0,3 | 28,3 ± 0,7 | 29,1 ± 0,4 |
| CT(°C) March 2015 | 28,9 ± 0,3 | 29,3 ± 0,3 | 29,5 ± 0,4 |

and showed that 70 m depth values measured during our survey corresponded well with satellite winter data (SST as an example below)

|  | WMA | EMA | EGY |
|---|---|---|---|
| SST(°C) July 2014 | 24,9 ± 0,2 | 24,2 ± 0,7 | 26,5 ± 0,2 |
| $CT_{70\,m}$(°C) March 2015 | 25,3 ± 0,3 | 24,8 ± 0,9 | 26,1 ± 0,9 |

Of course, this is an extrapolation, but we also showed a good agreement between chlorophyll a (Chl a and SSChl a), and for carbonate variables (DIC, Alk, $pCO_2$) measured during OUTPACE at 70 m depth and the same variables provided at the ocean surface in winter by the climatology study of Takahashi et al. (2014): see paper for details. Considering that 5 different available winter variables gave a good agreement, we consider that we can use the proposed extrapolation for all other variables to obtain our 'first order' budgets. Of course, this is the simplest 'model' to obtain seasonal variations in the absence of seasonal data, but highlighting the importance of seasonal variations in areas where they are generally considered as unimportant, or even non-existent, is an important issue in our opinion.

**These are minimum estimates at best given that in almost all profiles the chl-a maximum is below 70m.**

Yes, but the DCM depth in oligotrophic areas is essentially related to a maximum of pigments (chlorophyll a) by carbon unit and did not correspond to a maximum in biomass (see figure 5g) or in production (see figure 6a). Almost all C-biomass and C-production were measured in the upper 0-70 m, even if the DCM depths were below 70 m.

**2. How can matter, recovered from sediment traps, be taken at face value and trap biases be not considered? Or, even better, corrected for! This issue has been extensively discussed: Gardener 2000 in The changing ocean carbon cycle, Hanson, Ducklow, Field, eds., Cambridge Univ. Press, p 240-281; Buesseler et al. 2007, J.Mar.Res. 65, 345-416).**

We apologize for the lack of precision in the description of our sediment trap data methodology. We followed most of the major recommendations proposed by Gardner (2000) and Buesseler et al (2007) but not the [234]Th calibration. Nevertheless, we recognize that our protocol was not detailed enough, and we propose some modifications (see below). Gardner (2000) explain that [234]Th calibration was not done around HOT because the scavenging of [234]Th is so low in oligotrophic regions that this calculation is subject to major errors. We therefore assume that Referee #1 does not want us to explain why we did not use the [234]Th calibration in our oligotrophic study area. Concerning the 3 major biases in Buesseler et al. (2007):

  i) Hydrodynamics: please note that for our quasi-Lagrangian study, we sampled in low horizontal advection areas and chose regions with low surface currents on purpose (de Verneil et al., this issue). Buesseler et al. (2007) concluded from several experiments in Bermuda that at low velocities, there may not be a strong hydrodynamic effect on the mass flux measured with surface-tethered drifting traps. We have no isotope measurements to quantify this effect, but we can argue that our sample conditions should minimize this bias.

ii) Zooplankton swimmers: according to Buesseler (2007), the term 'swimmer' refers to metazoan zooplankton (and occasionally small fish) that are thought to actively enter sediment traps. Their "uninvited presence in the sediment trap" means that swimmers may be seen as a problem, and solutions are proposed to avoid it. Although Buesseler et al. (2007) encourages the continual development of swimmer avoidance traps in order to improve estimates of in situ particle flux, they explain that screening is often used for deep sediment traps as many of the swimmers are large and can be readily removed using screens. Since swimmers caught in oligotrophic environments tend to be smaller, however, traps used there will require sorting and manual removing. As the method for swimmer removal varies widely (Buesseler et al., 2007), we understand the need to describe it precisely and we propose to do so. Please note that for French cruises, the same person is in charge for the whole analysis and protocol development. Particularly for the removing of swimmers, and even if the method is subjective, it allows for a direct comparison of all measurements.

Iii) Solubilization: This point is discussed below.

**Solubilization into collection cup supernatants is (almost) not considered at all but this is important especially in shallow traps, see Kähler and Bauerfeind (2001), Limnol. Oceanogr. 46, 719-723; Antia (2005) Biogeosci. Discussions 275-302. In the case of phosphorus, supernatants have been measured but assigned 100% to swimmers! What about P in the passive flux?**

According to Antia (2005, introduction, paragraph 4) cited by Referee #1, they worked with traps below 500 m depth: "..shallow deployments for merely hours to days in high-swimmer environments would require a different approach, since swimmers are a major source of dissolved elements in such trap samples, and there is no reliable way to separate swimmers contribution of different elements from that originating from the passive flux". We do indeed consider that swimmers are probably the major source of dissolved P, and we agree that there is no reliable way to separate swimmers' contribution of different elements from that originating from the passive flux. Furthermore, using formaldehyde precluded DOC measurements. Therefore, it will not be possible to correct for this bias. This bias will be mentioned in the new version of the ms.

**DOC could obviously not be measured (formalin) and DON was not, or is not reported. Hence almost all P and much N and C "export" by particles is missed in the trap data.**

Yes, the only dissolved contribution measured, because it was supposed to be the largest (as shown by Antia (2005)), was the P contribution. Because we prefer not to enter into the details considering this controversial point, which is outside the main focus of our paper, we will add this methodological issue as another hypothesis to possibly explain part of the unbalanced carbon budgets.

**Also one of the traps is used in the balances of two regions (Fig. 7).**
Yes, the trap deployed at LD A for several days was considered for the Melanesian archipelago (WMA and EMA) and the trap deployed at LD C for several days was considered for WGY. We understand Referee #1's remark, but this point was clearly explained in the ms and without specific trap data, we considered that it would be worse to withdraw the EMA budget.

3. **Zooplankton vertical migration is conjectured to be the one important export mechanism with only implied evidence (ADCP data not shown).**

Zooplankton and micronekton diel vertical migrations have already been observed and considered to play a significant role in the WTSP (Smeti et al., 2015), a reference that we missed in our ms and propose to add in the revised version. The ADCP data (figures below) show the vertical migrations at LDA.

Smeti H., Pagano M., Menkès C., Lebourges Dhaussy A., Hunt B. P. V., Allain V., Rodier M., de Boissieu F., Kestenare E., Sammari C.: Spatial and temporal variability of zooplankton off New Caledonia (Southwestern Pacific) from acoustics and net measurements. J. of Geophys. Res. Oceans, 120 (4), 2676-2700. ISSN 2169-9275, 2015.

[Figure]

[Figure]

Raw SADCP Echo time series data at LDA station 38 KHz (above) and 150 KHz (below) showing diel vertical migrations of zooplankton-micronecton during the 5 days of the station occupation.

**But to present swimmers attracted to the sediment trap as evidence of vertical migration ("Zoo/Particulate mass flux ratio" in Tab.4) or "mean carbon export by swimmers" (p.13 line 23) is downright nonsense. Swimmers killed in the trap cannot move upward again. And of what should swimmers be representative? Certainly not of a vertical flux.**

We understand the problem. "mean carbon export by swimmers" (p.13 line 23) is definitely not the best term because it implies that this carbon is definitely exported from the upper layer, while most vertical migrators also ascend to the surface waters as part of the diel cycle (this explains why they should be removed (Buesseler et al., 2007)). This sentence and the entire paragraph will be removed. Moreover, and because swimmers may be attracted to the sediment trap, we will only speak about zooplankton content and not fluxes.

**4. New production and primary production are not neatly distinguished in the text (e.g. p.13 line37; p13 line 19). Most PP is recycled in these environments thus PP cannot explain the drawdown of DIC.**

New production is the production associated with new N (Nitrate from below and from nitrogen fixation), as clearly mentioned by Dugdale and Goering (1967), even if most people forget nitrogen fixation as a significant new N source. Rates of nitrogen fixation are found to be high or higher than nitrate uptake, in some cases suggesting an important role for nitrogen-fixing phytoplankton (Dugdale and Goering, 1967). As mentioned by Referee #1, it is clearly a fraction of the whole primary production. Even if most of the PP is recycled, the e-ratio is higher than in other oligotrophic areas less influenced by $N_2$ fixation (see paper).

**p. 13 line 19 : The sentence is : « Considering that half of this loss (12.5,%) happens at 500 m depth following ingestion of the water column's whole PP(new biomass) in the upper layer, … »**

This sentence will be removed in the new version of the ms.

**p.13 line37 : The really high N$_2$ fixation rates in the MA, compared to other areas in the world (Bonnet et al., 2017), may provide the nitrogen required for primary production, creating the necessary decrease in p$_{CO2}^{oc}$ to stimulate CO$_2$ invasion.**

This sentence will be replaced by: The really high N$_2$ fixation rates in the MA, compared to other areas in the world (Bonnet et al., 2017), may provide the new nitrogen required for new production, creating the necessary decrease in p$_{CO2}^{oc}$ to stimulate CO$_2$ invasion.

**5. The ms. was to me difficult to read because of abounding abreviations and numbers in the text, repeated checks were necessary whether the authors refer to their own data or speculate. No story is told convincingly. But my rejection of the ms. is mainly based on points 2 and 3 above.**

We recognize that the ms is not easy to read, but as mentioned by Referee #1, the ms deals with an extensive dataset, and we think that it is the main reason for this difficulty. Referee #1 recommended rejecting the ms mainly on the basis of points 2 and 3 above. We hope he will reconsider his decision in view of the detailed responses given here and the near-deletion of this part, which is of secondary importance in terms of the main focus of our ms. The role of zooplankton and vertical migrations in this region will be the subject of a new paper by C. Menkes and collaborators in the near future. We now only suggest, in a section of the Discussion and among 2 other hypotheses, a possible role of diel vertical migrations of zooplankton in the transfer of carbon.

\* Protocol in the new version

*Settling particulate matter mass and C, N, and P flux measurements*

The settling of particles in the water column outside of the upper layer was measured using 2 PPS5 sediment traps (1 m$^2$ surface collection, Technicap, France) deployed for 4 days at 150 and 330 m at LD A (MA) and LD C (WGY) stations (Fig. 1). The PPS5 traps are covered with baffled lids (mesh 1 cm$^2$) to reduce current shear at the mouth of the trap, but also to prevent large zooplankton and fish from entering the traps. Particle export was recovered in polyethylene flasks screwed on a rotary disk which automatically changed the flask every 24-h to obtain daily material recovery. The flasks were previously filled with a 2% (v/v) buffered solution of formaldehyde (final pH≈8) prepared with in situ deep seawater.  A sample of this water is kept to measure dissolved nutrients (phosphate and silicate). Immediately after trap retrieval, samples were stored at 4 °C in the dark until they were processed. Back in the laboratory, one part of the sample's supernatant was kept and stored at 4°C to measure dissolved nutrients (phosphate and silicate), and pH was checked on every trap sample. Swimmers (all organisms deemed to have actively entered the trap) were identified under a stereomicroscope and carefully removed with plastic fine-tipped forceps and placed into small vials with some of the reserved trap preservative. The main species removed were copepods, crustaceans (ostracods, euphausiids, amphipods) and pteropods. Microphotographs of each sample were taken. After the swimmers were removed, the whole sample was then rinsed 3 times with ultrapure (MilliQ) water in order to remove the salt and then freeze-dried. Mass particle fluxes were obtained by weighing the freeze-dried sample 5 times. The accuracy of the weighing (and thus of the flux) was 1 % over the whole data series. In this study, swimmers were rinsed and freeze-dried and their dry weight was also determined.  Settling particulate matter and swimmers were analyzed separately on an Elemental Analyzer coupled to an Isotope Ratio Mass Spectrometer EA-IRMS (Integra2, Sercon

Ltd) to quantify total C and N. Total P was analyzed as described in Sect. 2.2. The total element measurements for the settling particulate matter were considered to represent the settling particulate organic C, N, P. The results are presented Sect. 2.2 (Table 4).

---

## Author Comment (AC2) · 19 Mar 2018

**This manuscript reports upper water column biogeochemical observations from a cruise in the western tropical South Pacific. The cruise programme had a particularly novel finding: elevated rates of nitrogen fixation throughout the Melanesian Archipelago. These results have already been reported: briefly in Bonnet et al. (2017), and in detail in a companion manuscript as part of the cruise special issue in Biogeosciences Discussions (Bonnet et al., in review). The novel contribution of the present manuscript is that an upper water column carbon budget for the region, built from observations on the cruise and longer time series datasets, appears to require an additional fixed nitrogen source other than seasonal entrainment of waters below the mixed layer.**
**They interpret this deficit as originating from the enhanced nitrogen fixation observed (Bonnet et al., 2017; Bonnet et al., in review). Provided all carbon system calculations are correct which is not my area of expertise I think the manuscript is of significant interest to the scientific community. I do however have a number of comments that should be addressed.**

We thank Referee #2 for remarking on the novel findings of the cruise program and the significant interest for the scientific community of our ms. In the following paragraphs, the original review comments from Referee #2 are in bold and our responses are interspersed in normal characters. The main conclusion of our paper is that the discrepancies between the depth of the nitracline and phosphacline together with evidence of deeper mixing at 70 m allows excess P to annually reach the upper surface, support nitrogen fixation, and induce a significant biological soft tissue pump in the MA. Calculations of the carbon system have been done with the seacarb R package following all instructions provided by the authors of the package and the manuscript has been read by a carbon chemistry specialist (Nicolas Metzl, as mentioned in the Acknowledgements).

Main comments:
**Clarity of writing. The manuscript requires a large number of corrections for English, sentence structure, and correct terminology. This will much improve the readability of the paper and make it more readily understandable. There were a number of instances where I had to read a paragraph more than once to work out what message the author was trying to convey (and not always with 100% success!). Therefore, I would recommend going through the paper in detail with a native English scientist in order to check sentence structure and terminology for simplicity and clarity. I have made some suggestions in the specific comments section, but these are only examples; there are more throughout the manuscript.**

We will follow the recommendations provided and we will send the revised ms to a native English speaker for proof-reading.

**The role of Fe. The manuscript argues for a primary role of Fe in enabling nitrogen fixers to become established in the Melanesian Archipelago, but then invoke phosphorus availability as the primary factor controlling the activity of nitrogen fixers. Accordingly, Fe is frequently stated as 'high' (or 'Fe-replete') in Melanesian Archipelago area.**
**Where are the Fe data? The authors do give some values from a paper in review that appear high, however, since this is central to the manuscripts conclusions, some more details are needed to describe how Fe concentrations varied though surface waters of the three regions discussed. Surface Fe supply from shallow hydrothermal vents is also mentioned, which is very interesting, however as I understand it none of these data are yet published. Regardless, whilst this could significantly influence the overall Fe budget and capacity of the Melanesian Archipelago system for nitrogen fixation, this does not preclude periodically or seasonally low levels of Fe in surface waters of the Melanesian Archipelago. In other words, what evidence is there that Fe is at high, steady state value in this biogeochemical province? Supplied Fe has a short lifetime in seawater because of rapid scavenging and concentrations following a point source supply diminish rapidly**

**and require continuous inputs to lead to sustained high surface concentrations (such as under the Saharan dust plume in the tropical North Atlantic). Other Fe values reported by Moisander et al. (2011) in the vicinity of the region appear lower than the two values stated.**

After re-stating our main conclusion regarding the role of iron in this area, several important questions were raised by Referee #2 that need individual responses.

Where are the Fe data? We understand this to mean 'Where are the OUTPACE Fe data?', because other values are given clearly indicating the large difference between the gyre (~0.1 nM measured in the upper 350 m water column of the SP gyre (Blain et al., 2008)) and the MA (0.57 nM reported by Campbell (reference in the ms) as an example). We reported only the average concentrations in the photic layer measured during OUTPACE (Guieu et al., in revision), which confirmed this difference in iron availability and was sufficient for our purpose: 1.7 nM in the MA compared to 0.3 nM in the WGY. As requested by Referee #2, we will add the averaged 0-70 m integrated concentration corresponding to our 3 areas in the revised ms: 0.57±0.14 nM (WMA); 1.18±1.02 nM (EMA); 0.28±0.03 nM (EGY), but we cannot present detailed data that will be published in Guieu et al. paper. elsewhere!

We appreciate Referee #2's interest in the role of hydrothermal Fe sources, which is really an important point. The paper by Guieu et al. is currently under revision (Nature Scientific Report) and will we hope be accepted soon. We previously highlighted the importance of shallow hydrothermal sources in providing iron to the MA: following the South Equatorial Current from east to west (see our figure in Bonnet et al., PNAS, 2017), water flows across the most active volcanic area of the world ocean (the low bathymetry area in Figure 1) and becomes enriched in iron, as reported by several authors.

Referee #2 indicates that this does not preclude periodically or seasonally low levels of Fe in surface waters of the Melanesian Archipelago, and Referee #2 is right. Nevertheless, if we consider iron as a nutrient-type trace element that undergoes seasonal variations, the lowest surface concentrations should be during summer stratification (especially if the source of iron is from below). The OUTPACE measurements are from the summer, and we still measured high Fe concentrations. Therefore, even if we have no evidence that Fe is at a high, steady-state value in this biogeochemical province, we can reasonably hypothesize that high concentrations are expected all year long. Because data will always be better than a long explanation, we contacted Martine Rodier (cited in the ms). To the best of our knowledge, she is the only person who has measured seasonal iron data in the WTSP. She kindly agreed to send us these data to answer Referee #2's comment. Surface (0-60 m) average iron concentrations measured during the DIAPALIS cruises (http://www.obs-vlfr.fr/proof/vt/op/ec/diapazon/dia.htm) in the open ocean east of New Caledonia using the method detailed in Blain et al. (2007) were [May 2002 : 1.00 ± 0.35 nM (n=10) ; Feb. 2003 : 1.57 ± 0.63 nM (n=6) ; June 2003 : 1.75 ± 0.53 nM (n=6) ; Oct. 2003 : 1.65 ± 0.86 nM (n=5)], indicating higher concentrations than in the gyre and no clear seasonal variations. During the same period, DIP turnover times varied from months to several hours (Van den Broeck et al., 2004; Moutin et al., 2005: references in the ms).

The last part concerns the lifetime of iron in seawater. How can iron concentrations be considerably above the iron hydroxide solubility in seawater considered to be < 0.1 nM (Liu and Millero, 2002)? Even if we had no direct measurements during OUTPACE, it is now accepted that organic ligands, some of them probably also originating from hydrothermal sources, allow higher solubility (and also longer lifetime) of iron than initially expected. All this will be discussed in the Guieu et al. paper, but please note that Fitzimmons et al. (2014) already explained that "dFe must have been transported thousands of kilometers away from its vent site to reach our sampling station", a reference that will be added in the revised version.

The DFe concentrations reported by Moisander et al. (2011), while lower than the average concentrations reported for the photic zone by Guieu et al. (in revision), were at least twice higher than the ~0.1 nmol L-1 measured in the upper 350 m water column of the SP gyre (Blain et al., 2008). Thus, these data still indicate higher concentrations closer to the MA. Furthermore, phosphate

concentration (SRP), except at their station 17, was largely above our measurements and specific conditions, different from those observed during the OUTPACE cruise, may have been observed. Again, a detailed, seasonal survey would be of great interest in this region, which is one of the perspectives suggested in the present paper.

Fitzsimmons, J.N., Boyle, E.A., and Jenkins, W.J.: Distal transport of dissolved hydrothermal iron in the deep South Pacific Ocean. Proc Natl Acad Sci U S A. 2014 Nov 25;111(47):16654-61. doi: 10.1073/pnas.1418778111, 2014.
Liu, X., and Millero, F.J.: The solubility of iron in seawater. Mar. Chem. 77, 43-54, 2002.

**Sources of fixed nitrogen. Currently the authors use profiles of nitrate concentrations measured during their cruise and climatology of MLD to estimate nitrate entrainment during deeper wintertime mixing. From this they conclude that nitrate input to the surface mixed layer by this process is minimal. Indeed, looking at the depth profiles of nitrate (nitracline _70m depth) and MLD climatology (maximum mixed layer of _75m) this would appear to be sound. However, in using climatological average mixed layer depths, the authors do not consider periodic entrainment of much deeper waters with much higher nitrate concentrations by transient storms (see, for example, general cyclone passages at: https://en.wikipedia.org/wiki/Tropical_cyclogenesis#/media/File:Global_tropical_cyclone_trackse dit2.jpg). What is the role of periodic nitrate entrainment by these deep mixing events that are not characterised by average MLD climatology? It is worth noting that such storms would also entrain DIC into the mixed layer, as well as nitrate, and the net effect on carbon budgets might be low.**

Referee #2 asked about the effect of cyclones on nitrate entrainment. We thank Referee #2 for this comment, and the paper by Law et al. (2011) in the North Tasman Sea shows precisely the effect of such a cyclone. There was in fact no nitrate entrainment but a phosphate entrainment due to explicit differences in nitracline and phosphacline depths, and this allowed nitrogen fixation to be enhanced in a process close to the one we describe in our ms, but for physical forcing acting at a much finer horizontal spatial scale than the winter vertical mixing. Yes, a cyclone may entrain nitrate from below or enhance nitrogen fixation through introduction of phosphate, and therefore enhance the biological soft tissue pump. Incidentally, during OUTPACE, there was a strong wind forcing event within the study region. Cyclone Pam entered the Southwest Pacific in early March, and a drop in SST and increase in Chl followed in its wake (see figures provided by the response to Referees for another article in this special issue, de Verneil et al., 2017, page 8: https://www.biogeosciences-discuss.net/bg-2017-84/bg-2017-84-AC2-supplement.pdf). The storm did indeed have a fertilizing effect but at relatively short spatial (around Vanuatu islands) and time (around 2 weeks) scales, compared to the larger-scale processes highlighted in the present study. In any case, this additional effect and the paper by Law et al. (2011) will be added in the revised version of the ms, mainly to explain that while episodic storms such as Cyclone Pam can induce deep mixing, which in turn brings phosphate (but no nitrate, i.e. excess P) to the surface and enhances nitrogen fixation, this mechanism may concern limited spatial and temporal scales compared to winter mixing.

Law, C. S., Woodward, E. M. S., Ellwood, M. J., Marriner, A., Bury, S. J., and Safi, K. A.: Response of surface nutrient inventories and nitrogen fixation to a tropical cyclone in the southwest Pacific. Limnology and Oceanography 56, 1372–1385, 2011.

**What is the mesoscale eddy activity in this region? These have been shown to supply significant N to other oligotrophic waters. Could eddies be supplying additional nitrate? See, for example, Falkowski et al. 1991; Oschlies and Garcon, 1998.**

According to de Verneil et al. (this issue; see again the supplemental .pdf referred to above, pgs. 4-5), mesoscale vertical fluxes due to quasi-geostrophic forcing calculated from satellite data were weak and would act on a layer displaced from the relevant nutrient reservoirs. The mesoscale activity was considered outside the scope of this large-scale study, and is studied in other papers in this special issue, such as Rousselet et al., (in revision). The idea was not to ignore mesoscale activity but rather to demonstrate that interesting results can be obtained from considering larger spatial and temporal scales.

**Additional sediment trap sample details. Some details with regards to the type of material found in the sediment traps could be valuable to support the 'mechanistic' discussion with respect to nitrogen fixation-fuelled carbon export. Were significant Trichodesmium colonies found in the sediment traps? Or diazotrophic diatoms? Or zoo-plankton? Partly related to this, are there any details about phytoplankton community structure determined from HPLC pigment analyses, flow cyctometry, or microscopy? I appreciate that all these details could be in another article in the special issue, but I cannot see it mentioned. More generally the manuscript should be very clear/explicit as to what the new data in this manuscript are, and what is has been published/is in review elsewhere. XX**

As requested by Referee #2, we have added sediment trap sample details in the new version of the ms. The protocol has been rewritten, as follows:

*Settling particulate matter mass and C, N, and P flux measurements*

The settling of particles in the water column outside of the upper layer was measured using 2 PPS5 sediment traps (1 $m^2$ surface collection, Technicap, France) deployed for 4 days at 150 and 330 m at LD A (MA) and LD C (WGY) stations (Fig. 1). The PPS5 traps are covered with baffled lids (mesh 1cm$^2$) to reduce current shear at the mouth of the trap, but also to prevent large zooplankton and fish from entering the traps. Particle export was recovered in polyethylene flasks screwed on a rotary disk which automatically changed the flask every 24-h to obtain daily material recovery. The flasks were previously filled with a 2% (v/v) buffered solution of formaldehyde (final pH≈8) prepared with in situ deep seawater. A sample of this water is kept to measure dissolved nutrients (phosphate and silicate). Immediately after trap retrieval, samples were stored at 4 °C in the dark until they were processed. Back in the laboratory, one part of the sample's supernatant was kept and stored at 4°C to measure dissolved nutrients (phosphate and silicate), and pH was checked on every trap sample. Swimmers (all organisms deemed to have actively entered the trap) were identified under a stereomicroscope and carefully removed with plastic fine-tipped forceps and placed into small vials with some of the reserved trap preservative. The main species removed were copepods, crustaceans (ostracods, euphausiids, amphipods) and pteropods. Microphotographs of each sample were taken. After the swimmers were removed, the whole sample was then rinsed 3 times with ultrapure (MilliQ) water in order to remove the salt and then freeze-dried. Mass particle fluxes were obtained by weighing the freeze-dried sample 5 times. The accuracy of the weighing (and thus of the flux) was 1 % over the whole data series. In this study, swimmers were rinsed and freeze-dried and their dry weight was also determined. Settling particulate matter and swimmers were analyzed separately on an Elemental Analyzer coupled to an Isotope Ratio Mass Spectrometer EA-IRMS (Integra2, Sercon Ltd) to quantify total C and N. Total P was analyzed as described in Sect. 2.2. The total element measurements for the settling particulate matter were considered to represent the settling particulate organic C, N, P. The results are presented Sect. 2.2 (Table 4).

Papers specifically dealing with trap material and nitrogen budgets are cited (Caffin et al., this issue, Knapp et al., this issue) to confirm the preponderant role of nitrogen fixation in the MA. Zooplankton

(swimmers) C, N, P contents in the traps were already presented in the ms but not the species, although this information is available on our website upon request. The title of the special issue is: "Interactions between planktonic organisms and biogeochemical cycles across trophic and $N_2$ fixation gradients in the western tropical South Pacific Ocean: a multidisciplinary approach (OUTPACE experiment)", and therefore most of the papers deal with biology and biogeochemical cycles. The aim of this paper was to focus on large scale and biogeochemical budgets only. Our attempt to demonstrate the influence of zooplankton migration as a mechanism within our biogeochemical budgets had its own limitations (in addition to Referee #2's comments, see Referee #1). As a result, our subsequent decision has been to focus on the main, biogeochemical budget message and to shorten the Discussion.

The objective of the OUTPACE special issue is to provide an original opportunity for a group of researchers to focus on the "Interactions between planktonic organisms and biogeochemical cycles across trophic and $N_2$ fixation gradients in the western tropical South Pacific Ocean". The results will be published in a single book at the end. It is a multidisciplinary approach with a tight time schedule, and the major objective is to share data between project collaborators to give the better analysis of the dense datasets. Data may be used several times in different papers of the special issue focusing on different scientific questions. In this case, the method is described in full detail only in one paper that is clearly referenced as the prevalent one for the method. In our paper, we focus on the C, N, P inventories and fluxes in the upper 0-200 m layer, but as an example the $N_2$ fixation rates were also presented in a more specific paper presenting the methodology in detail.

**More specific comments**
**ADCP data. These data are mentioned with regards to the zooplankton migration, with implications for sub-mixed layer carbon export. Can we see plots of the data?**

[Figure]

[Figure]

Raw SADCP Echo time series data at LDA station 38 KHz (above) and 150 KHz (below) showing diel vertical migrations of zooplankton-micronecton during the 5 days of the station occupation.

Please note that the diel vertical migration of zooplankton-micronecton will now be detailed in another paper by Menkes et al. (in prep) who have already worked and published data on this subject (Smeti et al., 2014), a reference that we will add in our Discussion.

Smeti H., Pagano M., Menkès C., Lebourges Dhaussy A., Hunt B. P. V., Allain V., Rodier M., de Boissieu F., Kestenare E., Sammari C.: Spatial and temporal variability of zooplankton off New Caledonia (Southwestern Pacific) from acoustics and net measurements. J. of Geophys. Res. Oceans, 120 (4), 2676-2700. ISSN 2169-9275, 2015.

**How were the reported zooplankton respiratory carbon losses to the sub-mixed layer calculated?**

We used the quota of primary production (PP) proposed by Pagano and Ikeda (cited reference) to estimate carbon losses by respiration considering that all daily PP was grazed by zooplankton. In any case, note that this part has been removed following the remarks by Referee #1.

**DOP and alkaline phosphatase activity. Figure 5 shows significant phosphate available as DOP, even in DIP-depleted Melanesian Archipelago waters. Could the P demand of nitrogen fixers be sustained by DOP access?**

Yes, in theory, but it is still not clear if it is a direct uptake of DOP molecules or rather a DOP to DIP transformation outside of the cells (ecto-enzyme activity) followed by DIP assimilation. In any case, the turnover times of DIP are really short in the MA, suggesting both rapid recycling and really low DIP concentrations. One of the interesting results we provided is a seasonal increase in DOP, suggesting a relatively low utilization of the major part at this time scale.

**Was alkaline phosphatase activity measured? Recent findings show that Fe/Zn is required for some dominant forms of alkaline phosphatase and so Fe/Zn availability could also control P access, in addition to N access via nitrogen fixation (Mahaffey et al., 2014; Browning et al., 2017i; Landolfi et al., 2015).**

Yes, alkaline phosphatase activity was measured by F. Van Wambeke. Data show higher Vmax in the MA compared to the GY. Referee #2 indicates an interesting point concerning Fe/Zn availability and a possible control of alkaline phosphatase activity. Unfortunately, we do not have Zn measurements at the present time.

**Relative influence of iron and nitrogen in the WGY region. Biologically accessible nitrogen is inferred to be the limiting resource to the overall phytoplankton community in the WGY (western gyre region) (with Fe being the limiting nutrient for diazotrophs). What is the evidence that nitrogen and iron are not directly co-liming to the overall phytoplankton community in this region (Saito et al., 2014; Browning et al., 2017) both studies through the edge of gyre boundaries)? In the study of Moisander et al. (2011), which was in proximity to the OUTPACE cruise region, a bottle enrichment experiment displayed Fe was serially limiting (following nitrogen) to the overall phytoplankton community, supporting this possibility.**
**Co-limitation. Co-limitation is not mentioned once in the manuscript but it seems central to the discussion. For instance, iron and phosphorus could co-limit nitrogen fixation (Mills et al., 2004), but iron could also co-limit phosphorus acquisition by the microbial community from the DOP pool, and iron and nitrogen could co-limit the overall phytoplankton community. These interactions and potential for feedbacks are pertinent when hypothesising a role of 'ultimate' limiting nutrients and potential for future change (both topics of this manuscript). To quote Moore et al. (2013; referenced in the manuscript): "Establishing the identity of a single ultimate limiting nutrient may thus be less relevant than understanding the controls on, and feedbacks pertaining to, any given process".**
**Personally I would avoid conclusions/discussion that heavily refer to 'ultimate' limiting nutrients (e.g. Moore et al., 2013; Tyrrell, 1999) as this is more relevant to larger spatial-temporal scales than this dataset can be used for.**

It is a very important point that needs to be discussed because as Referee #2 said, it is central to our discussion and even part of the title.

The introduction of Moutin et al. (2005) was as follows: "In a nitrogen- limited ocean, the input of 'new' nitrogen (i.e. not related to organic matter recycling) into the photic zone controls primary production (Codispoti, 1989). Nitrogen fixation in the ocean is a source of 'new' nitrogen. Thus, a fundamental question arises as to: 'What factors control nitrogen fixation in the ocean?' What are the factors that control N2 fixation over annual or longer time-scales (Falkowski 1997, Tyrell 1999, Letelier & Karl 1998, Tyrell 1999) and are these distinct from 'physiological' factors that may temporarily control the process of nitrogen fixation? Light, temperature (Carpenter et al. 2004) and nutrient availability, particularly that of phosphate (Sañudo-Wilhelmy et al. 2001, Mulholland et al. 2002, Fu & Bell 2003) and iron (Behrenfeld & Kobler 1999, Kustka 2002), could physiologically control the kinetics of nitrogen fixation. These factors could be different from the 'systemic' factor (Paasche & Erga 1988) that controls the cumulative biomass over time within a particular oceanic area, and ultimately, when considering all the oceanic provinces, the amount of nitrogen introduced via di-nitrogen fixation in the world's oceans.

Physiological factors can be investigated using short-term experiments such as selective enrichment experiments showing that there may be co-limitation of diazotrophs by both iron and dissolved inorganic phosphate (DIP) in certain situations (Mills et al. 2004). However, such short-term limitation may not control accumulation of diazotroph biomass over time. For example, if the 'systemic' limiting factor is DIP availability in a particular area, a more or less high iron availability will only drive the system to a more or less rapid consumption of DIP. The cumulative biomass of Trichodesmium spp., which depends essentially on the DIP consumption, will not be affected in the long term by iron availability. Considering the cumulative biomass as the end product of the nitrogen fixation process, 'physiological' factors act as catalytic factors only. Thus, knowing the systemic controlling factor is of prime necessity and can only be assessed using annual or longer- term studies on nutrient availability and uptake kinetics parameters of the species studied."

Yes, nitrogen is the first limiting nutrient, rapidly followed by others in the WTSP using short term experiments (Van Wambeke et al., this issue), but 5-10 days were necessary to obtain an increase in PP, chl a and export production after a DIP enrichment in a minicosm experiment, suggesting that classical methods (short-term microcosm experiments) used to quantify nutrient limitations of PP may not be relevant (Gimenez et al., 2016), at least in the WTSP.

As high nutrient concentrations in High Nutrient Low Chlorophyll (HNLC) areas (Minas et al. 1986) may be considered as the result of an inefficient biological carbon pump (Sarmiento and Grüber, 2006), high or relatively high phosphate concentrations (and high DIP turnover time) in the south Pacific gyre (Moutin et al., 2008; Moutin et al., this issue) may be the result of inefficient $N_2$ fixation. Conversely, the low P availability (low concentration and DIP turnover time) in the upper surface of the WTSP is the result of intense $N_2$ fixation.

Because iron concentrations are really low in the gyre and high in the MA (even during the strongest stratified period), and because of the specific iron needs of diazotrophs, iron availability is the best candidate for preventing nitrogen fixation in the gyre and allowing nitrogen fixation in the MA. Therefore, DFe may appear as the ultimate (systemic) nutrient control in the gyre and DIP may appear as the ultimate (systemic) nutrient control in the MA.

We hope we have managed to convince Referee #2 that we cannot avoid mentioning the term 'ultimate' in our large-scale study, and also that we cannot enter into a debate about co-limitation without specific data. We are only interested in the large scale nutrient limitation.

Gimenez, A., Baklouti, M., Bonnet, S., and Moutin, T.: Biogeochemical fluxes and fate of diazotroph-derived nitrogen in the food web after a phosphate enrichment: modeling of the VAHINE mesocosms experiment, Biogeosciences, 13, 5103-5120, https://doi.org/10.5194/bg-13-5103-2016, 2016.
Minas, H.J., Minas, M., and Packard, T.T.: Productivity in upwelling areas deduced from hydrographic and chemical fields. Limnol. Oceanogr., 31(6), 1182-1206, 1986.
Moutin, T., Van Den Broeck, N., Beker, B., Dupouy, C., Rimmelin, P., and LeBouteiller, A.: Phosphate availability controls Trichodesmium spp. biomass in the SW Pacific Ocean, Mar. Ecol. Prog. Ser., 297, 15-21, 2005.
Moutin, T., Karl, D. M., Duhamel, S., Rimmelin, P., Raimbault, P., Van Mooy, B. A. S., and Claustre, H.: Phosphate availability and the ultimate control of new nitrogen input by nitrogen fixation in the tropical Pacific Ocean, Biogeosciences, 5, 95-109, 2008.
Sarmiento, J. L., and Gruber, N.: Ocean Biogeochemical Dynamics, Princeton University Press, Princeton. 503 pp, 2006.

**Figures. Figures are generally clear but some details to captions need to be added.**
**For example, check that all vertical/horizontal lines in Figs 3–6 are defined in the figure caption (even if defined in the text). Figure 1 (map) is good for detail but not very good for placing the cruise in its wider geographic position. Could an inset map or similar with continents for geographic context be included in addition to the current map? See Bonnet et al. (in review in this special issue Fig 1).**

We thank Referee #2 for describing our figures as mostly clear. We have added the text corresponding to all vertical/horizontal lines in Figs 3–6 in the figure captions. We also add an inset map in Fig. 1: see below.

[Figure]

All comments indicated below have been taken into account in the revised version of the ms, unless a specific response is included within the Referee's comments. We thank Referee #2 for his/her careful reading of the ms.

**Tables. Currently commas rather than decimal points are used in tables. This could be confusing for some readers. Please change.**
**Spelling/sentence structure/grammar: see first main point, below are some examples:**
**Pg 2 Line 3: 'deep Sea' ! 'deep sea'**
**Pg 5 Line 41: allowing to draw first-order winter to summer seasonal variations ! 'allowing us to draw first-order winter to summer seasonal variations'**
**Pg 6 Line 10: 'and 2, 3, and 3' ! 'and 2, 3, and 4'?**

No, it is 'and 2, 3, and 3'

**Pg 8 line 23: 'Rapidly, seawater was collected in triplicates from the Niskin bottles in 2.3 L polycarbonates bottles at 6 depths (75 %, 54 %, 19 %, 10 %, 1 %, and 0.1 % surface irradiance levels), like for PP measurements' → 'As for PP measurements, seawater was rapidly collected in triplicate from the Niskin bottles in 2.3 L polycarbonates bottles at 6 depths (75 %, 54 %, 19 %, 10 %, 1 %, and 0.1 % surface irradiance levels).'**
**Pg 11, line 12: 'Otherwise, very low and improbable P contents were found in the swimmers' → Please be more specific with regards to 'very low' and 'improbable'**

We have added (see the previous column in Table 4) to the end of the sentence in order to be clearer. Please keep in mind that loss of P is known from living organisms after poisoning (Talarmin et al., 2011)

Talarmin, A., F. Van Wambeke, S. Duhamel, P. Catala, T. Moutin, and P. Lebaron. 2011. Improved methodology to measure taxon-specific phosphate uptake in live and unfiltered samples. Limnol. Oceanogr. Methods 9:443-453 (2011) | DOI: 10.4319/lom.2011.9.443.

**Pg. 11 line 37: 'large precipitation' → do you mean rainfall?**
**Pg 12 line 41: 'Else we need. . .' → 'Otherwise we need..'**
**Pg 18 Line 17: 'from an iron limitation in the east' → 'from probable iron limitation in the east'; currently no iron data is given to rule out other potential controls.**
**Pg. 18 Line 28: 'Furthermore, both diazotrophy and denitrification are known to undergo drastic alterations due to climate change.' → References needed to back up this statement?**

We have replaced 'known' by 'expected' and added 2 recent references (including other references), as requested:
McMahon, K. W., McCarthy, M. D., Sherwood, O. A., Larsen, T., and Guilderson, T. P.: Millennial-scale plankton regime shifts in the subtropical North Pacific Ocean, Science, 350, 1530-1533, 2015.
Lachkar, Z., Lévy, M., and Smith, S.: Intensification and deepening of the Arabian Sea oxygen minimum zone in response to increase in Indian monsoon wind intensity, Biogeosciences, 15, 159-186, https://doi.org/10.5194/bg-15-159-2018, 2018.

References
Browning, T.J., Achterberg, E.P., Yong, J.C., Rapp, I., Utermann, C., Engel, A. and Moore, C.M., 2017i. Iron limitation of microbial phosphorus acquisition in the tropical North Atlantic. Nature Communications, 8.
Browning, T.J., Achterberg, E.P., Rapp, I., Engel, A., Bertrand, E.M., Tagliabue, A. and Moore, C.M., 2017ii. Nutrient co-limitation at the boundary of an oceanic gyre. Nature, 551(7679), p.242.
Falkowski, P.G., Ziemann, D., Kolber, Z. and Bienfang, P.K., 1991. Role of eddy pumping in enhancing primary production in the ocean. Nature, 352(6330), p.55.
Landolfi, A., Koeve,W., Dietze, H., Kähler, P. and Oschlies, A., 2015. A new perspective on environmental controls of marine nitrogen fixation. Geophysical Research Letters, 42(11), pp.4482-4489.
Mahaffey, C., Reynolds, S., Davis, C.E. and Lohan, M.C., 2014. Alkaline phosphatase activity in the subtropical ocean: insights from nutrient, dust and trace metal addition experiments. Frontiers in Marine Science, 1, p.73.
Mills, M.M., Ridame, C., Davey, M., La Roche, J. and Geider, R.J., 2004. Iron and phosphorus co-limit nitrogen fixation in the eastern tropical North Atlantic. Nature, 429(6989), pp.292-294.
Moisander, P.H., Zhang, R., Boyle, E.A., Hewson, I., Montoya, J.P. and Zehr, J.P., 2012. Analogous nutrient limitations in unicellular diazotrophs and Prochlorococcus in the South Pacific Ocean. The ISME journal, 6(4), pp.733-744.

Moore, C.M., Mills, M.M., Achterberg, E.P., Geider, R.J., LaRoche, J., Lucas, M.I., McDonagh, E.L., Pan, X., Poulton, A.J., Rijkenberg, M.J. and Suggett, D.J., 2009. Large-scale distribution of Atlantic nitrogen fixation controlled by iron availability. Nature Geoscience, 2(12), pp.867-871.

Moore, J.K. and Doney, S.C., 2007. Iron availability limits the ocean nitrogen inventory stabilizing feedbacks between marine denitrification and nitrogen fixation. Global Biogeochemical Cycles, 21(2).

Saito, M.A., McIlvin, M.R., Moran, D.M., Goepfert, T.J., DiTullio, G.R., Post, A.F. and Lamborg, C.H., 2014. Multiple nutrient stresses at intersecting Pacific Ocean biomes detected by protein biomarkers. Science, 345(6201), pp.1173-1177.

Oschlies, A. and Garcon, V., 1998. Eddy-induced enhancement of primary production in a model of the North Atlantic Ocean. Nature, 394(6690), pp.266-269.

---

## Author Comment (AC3) · 19 Mar 2018

**Moutin et al.'s study investigates the regulation of the ocean C, N and P fluxes in the western tropical South Pacific Ocean, a key oligotrophic oceanic region for nitrogen fixation. They combine a large new dataset collected during the OUTPACE campaign (February-April 2015) with climatologies of upper water properties. The OUTPACE cruise followed a 4000 km longitudinal transect going from the Melanesian Archipelago (MA) to Papeete (French Polynesia). Comparing seasonal trends of C, N, and P mass balances for 3 main areas of the WTSP, they find that (1) the MA is a net sink of atmospheric CO2, mainly caused by the soft-tissue biological pump; (2) the MA biological pump results from both high rate of N2 fixation fuelling export production and mesozooplankton diel vertical migration; and (3) MA N2 fixation is essentially controlled by phosphate availability.**

**I think the tackled questions and presented results of this study are really interesting and important and deserve publication. There are lots of uncertainties on the role of nitrogen fixers in fuelling ocean production and export in oligotrophic area especially of the Pacific Ocean due to lack of observations. This region is particular important as covering a large area and presenting high rates of N2 fixation. I thus strongly recommend this work for publication. I have however the following main comments that need to be taken into consideration.**

We thank Referee #3 for remarking on the new large dataset obtained and his conclusion that the tackled questions and presented results are interesting, important and deserve publication. We really appreciate the detailed expression of our main conclusions, showing that the diel vertical migrations of zooplankton are only a part of one of the three main conclusions.
In the following paragraphs, the original review comments from Referee #3 are in bold and our responses are interspersed in normal characters.

Main comments

**Paper presentation - At the moment the results and the argument for the role of nitrogen fixation and mesozooplankton vertical migration for the atmospheric CO2 sink of the Melanesian Archipelago region are a bit convoluted. While it is nice to follow the steps of thought of the authors on how they come about to find these links, this is done a step too far: for instance, the second half of section 4.3 more or less repeats what is said in section 4.2. I recommend the authors to reorganise the results and discussion to be a lot more concise. This will help following the argument and strengthening the case of the paper.**

Following all the comments from the other Referees regarding zooplankton vertical migrations (which, as previously mentioned, was not the primary focus of this paper), and also the recommendation by Referee #3 regarding repetition in the Discussion, we have considerably shortened this part, deleting all sentences in the Abstract and Conclusion concerning this topic, and limiting the Discussion to what we consider to be our main focus. The diel vertical migrations of zooplankton-micronecton during OUTPACE will be the focus of a new paper by C. Menkes and collaborators in the near future.  With these changes, we hope to focus the reader's attention on what we consider to be the main message in this paper, the biogeochemical budgets.

**To help make a stronger case for the paper, I wonder also if it would be possible to add a diagram that visualises the main processes occurring in the region (N2 fixation, vertical migration of zooplankton, CO2 uptake . . .).**

We agree with Referee #3, but the diagram below has already been published in the OUTPACE preface paper (Moutin et al., 2017: https://www.biogeosciences.net/14/3207/2017/). Therefore, we proposed to add (see Moutin et al., 2017; their figure 1) after: The "soft tissue" pump in the new version of the ms. Please note that the vertical migrations of zooplankton were not specifically shown.

[Figure]

Figure 1. Major C fluxes for a biological pump budget and the main role of N₂ fixation. Biological pump is the C transfer into the ocean interior by biological processes. DIC is dissolved inorganic C, POC is particulate organic C, and DOC is dissolved inorganic C. See Moutin et al. (2012) for a detailed description.

Moutin, T., Doglioli, A. M., de Verneil, A., and Bonnet, S.: Preface: The Oligotrophy to the UlTra-oligotrophy PACific Experiment (OUTPACE cruise, 18 February to 3 April 2015), Biogeosciences, 14, 3207-3220, https://doi.org/10.5194/bg-14-3207-2017, 2017.

**Effect of zooplankton vertical migration - The current manuscript presents the vertical migration of mesozooplankton as an explanation for the missing sink in the upper C budget at WMA. It would be nice to quantify this process either based on other technique or by simply doing a mass balance (which the authors are shying away). MA net sink of atmospheric CO2 - It would be good to quantify the net sink in pCO2 and compare it with other estimates and with other regions.**

We now only suggest, in a section of the Discussion and among 2 other hypotheses, a probable role of diel vertical migrations of zooplankton in the transfer of carbon.

**Minor comments**

**P2, L3: Confusing sentence. In my mind mineral refers to carbonate and silicate which is at odd in association with the soft-tissue pump which refers to organic carbon.**

We have replaced 'mineral' by 'dissolved inorganic', to avoid confusion.

**P3, L35: Can you give slightly more information of the climatology of de Boyer- Montegut et al. (2004)? Mainly the type of collected data and coverage.**

The 2° resolution global climatology of the mixed layer depth (MLD) of de Boyer Montegut et al (2004) is constructed from nearly 5 million individual temperature profiles from the following databases: NODC WOD09 from 1941 to 2008, WOCE 3.0 from 1990 to 2002 and ARGO PFL from 1995 to sept. 2008. Profiles with any spurious data in the mixed layer have been removed (about 8% of the total profiles). For each selected profile, a MLD was estimated following the chosen criterion. In our study the MLD_DT02 data were used where the MLD is defined as the depth where the temperatures on the profiles differed from a fixed threshold of 0.2°C compared to the temperature at 10m. The MLD defined on individual profiles were gathered into monthly boxes of 2° latitude by 2° longitude.

For our study, pixels corresponding to each station of the different groups were extracted from the global climatology. Even if the number of existing profiles in the South Pacific Ocean is low, the MLD in the selected pixels were estimated from at least 5 and up to 80 profiles, depending on the pixel and month.

**P6, L39: Say in the Figure 2 caption, what the red lines refer to.**

We have added: "The vertical red lines indicate the period of the OUTPACE cruise:18 Feb. to 3 Apr. 2015." in the new version of the ms.

**P9, L27: Need to specify that shallow nitracline indicates oligotrophic waters.**

We have added: "with shallower nitracline depths » at the end of the sentence.

**P12, L35: Can you add the sum of the MA regions so that we can see that the MA area is a net sink of atmospheric CO2?**

We have rewritten this and added the mean atmospheric $CO_2$ input for the MA.

**P12, L40: Little bit misleading as the Table 6 and main text do not use the same unit. Can you add in the text the numbers in mol m-2 d-1 as well?**

We have modified the ms to use the same unit.

**P18, L40: Can you stipulate here if the source of P changes with climate change, how this might affect N2 fixation, zooplankton migration and CO2 sink?**

We did not think that the sources of P would change with climate change, but more precisely that the main expected alteration following climate change would strengthen the P limitation. This was developed in Moutin et al., (2008). In other words, the P-limited areas such as the MA, the Sargasso Sea or the Mediterranean Sea, might be extended.

Moutin, T., Karl, D. M., Duhamel, S., Rimmelin, P., Raimbault, P., Van Mooy, B. A. S., and Claustre, H.: Phosphate availability and the ultimate control of new nitrogen input by nitrogen fixation in the tropical Pacific Ocean, Biogeosciences, 5, 95-109, 2008.